# A novel ilarvirus protein CP-RT is expressed via stop codon readthrough and suppresses RDR6-dependent RNA silencing

**Nina Lukhovitskaya**[1]*, **Katherine Brown**[1], **Lei Hua**[2], **Adrienne E. Pate**[2], **John P. Carr**[2], **Andrew E. Firth**[1]*

**1** Department of Pathology, University of Cambridge, Cambridge, United Kingdom, **2** Department of Plant Sciences, University of Cambridge, Cambridge, United Kingdom

* nl358@cam.ac.uk (NL), aef24@cam.ac.uk (AEF)

**Data Availability Statement:** The AV2 sequences are available in GenBank with accession numbers OR820775, OR820776 and OR820777. The high throughput sequencing datasets generated and

## Abstract

Ilarviruses are a relatively understudied but important group of plant RNA viruses that includes a number of crop pathogens. Their genomes comprise three RNA segments encoding two replicase subunits, movement protein, coat protein (CP), and (in some ilarvirus subgroups) a protein that suppresses RNA silencing. Here we report that, in many ilarviruses, RNA3 encodes an additional protein (termed CP-RT) as a result of ribosomal readthrough of the CP stop codon into a short downstream readthrough (RT) ORF. Using asparagus virus 2 as a model, we find that CP-RT is expressed *in planta* where it functions as a weak suppressor of RNA silencing. CP-RT expression is essential for persistent systemic infection in leaves and shoot apical meristem. CP-RT function is dependent on a putative zinc-finger motif within RT. Replacing the asparagus virus 2 RT with the RT of an ilarvirus from a different subgroup restored the ability to establish persistent infection. These findings open up a new avenue for research on ilarvirus silencing suppression, persistent meristem invasion and vertical transmission.

## Author summary

Ilarviruses comprise a group of plant-infecting RNA viruses. Until now, ilarvirus genomes were thought to only encode two components of the viral replicase, a movement protein, the capsid protein, and the 2b suppressor of RNA silencing. Interestingly, whereas most plant viruses encode one or more proteins that suppress the host antiviral RNA silencing pathway, the gene encoding 2b is only present in some ilarviruses. Here, we identify an additional ilarvirus protein that is expressed via low efficiency ribosomal readthrough of the capsid protein (CP) stop codon. Readthrough ribosomes translate a short downstream readthrough (RT) domain to produce the capsid-readthrough fusion protein, CP-RT. We show that CP-RT is a second ilarvirus suppressor of silencing protein and is required for the model ilarvirus asparagus virus 2 to maintain persistent infection in the model plant *Nicotiana benthamiana*. These findings advance our understanding of ilarvirus molecular virology and virus-host interaction. Interestingly, there are substantial differences in the

analysed in this study have been deposited in the EBI ArrayExpress database under the accession numbers E-MTAB-13539. Raw data underlying the RT-qPCR graphs can be found at the qPCR_raw_data subdirectory of https://github.com/KatyBrown/ilarvirus_sequencing/.

**Funding:** A.E.F., N.L. and K.B. were supported by Wellcome Trust Senior Research Fellowships (106207/Z/14/Z, 220814/Z/20/Z) and a European Research Council grant (646891) to A.E.F. J.P.C. and A.E.P. were supported by a Leverhulme Trust grant (RPG-2022–134). L.H. was supported by a C4 Rice Project grant (INV-002970) from The Bill & Melinda Gates Foundation to the University of Oxford. The funders had no role in study design, data collection and analysis, decision to publish, or preparation of the manuscript.

**Competing interests:** The authors have declared that no competing interests exist.

RT peptide between divergent groups of ilarviruses and, in at least two cases, a separate RT peptide appears to have evolved independently. Thus it will be of future interest to investigate the function of CP-RT in different ilarvirus lineages.

## Introduction

RNA viruses often use non-canonical translation mechanisms to optimise the coding capacity of their compact genomes [1,2]. One such mechanism is stop codon readthrough, whereby a portion of ribosomes fail to terminate on the first stop codon but instead continue translation into the next in-frame stop codon to generate a protein with an alternative C-terminus. Various RNA viruses and retroviruses (including alphaviruses, tombusviruses, tobamoviruses and gammaretroviruses) utilise readthrough to control the expression level of the viral polymerase relative to other components of the replication complex. Several groups of plant positive-sense single-stranded RNA viruses (including luteoviruses, pomoviruses, benyviruses and furoviruses) use readthrough to synthesise a C-terminally extended coat protein. Some of these readthrough proteins have been shown to be important for virus transmission, systemic movement, virion formation or symptom development [3–8].

Traditionally, examples of stop codon readthrough have been classified into three types on the basis of stimulatory sequence elements [9,10,1]. Type I involves readthrough of a UAG stop codon and is stimulated by a CAR YYA hexanucleotide (R = purine, Y = pyrimidine) immediately 3′-adjacent to the UAG [11]. Type II involves readthrough of a UGA stop codon and is stimulated by 3′-adjacent nucleotides (generally CGG or CUA) and, in many cases, a downstream RNA stem-loop structure separated from the stop codon by approximately 6–12 nt. Type III is exemplified by the gammaretroviruses where it involves readthrough of a UAG stop codon stimulated by a 3′-adjacent purine rich octanucleotide followed by a compact pseudoknot [12]. In contrast, readthrough of a UAG stop codon in tombusviruses, luteoviruses and poleroviruses depends on base-pairing between a proximal RNA element and a distal RNA element often positioned hundreds or even thousands of nucleotides downstream [8,13–17]. An even more complex structure is involved in enamovirus readthrough [17]. Different readthrough signals have different efficiencies. For example, Loughran et al. [18] report 17% readthrough for the human *OPRL1* gene where readthrough is dependent on UGA CUA and more distal sequence elements, whereas UGA CUA on its own can stimulate readthrough at a level of ~1.5–2.9% [19,20]. Presumably each case of readthrough is fine-tuned through evolution to achieve the optimal expression level of the readthrough protein.

The plant virus genus *Ilarvirus* is classified in the family *Bromoviridae* and comprises several subgroups and many unclassified members (reviewed in Refs [21–23]). These viruses have spheroidal particles with *T* = 3 icosahedral symmetry. Ilarvirus genomes comprise three positive-sense single-stranded RNA molecules with 5′ cap structures but without 3′ poly(A) tails. The three RNAs are each packaged into separate virion particles. RNA1 encodes the 1a protein, a component of the viral replicase, with methyltransferase and helicase domains. RNA2 encodes the 2a protein, the second component of the viral replicase, with an RNA-dependent RNA polymerase domain. In ilarvirus subgroups 1 and 2, RNA2 has an additional 3′ ORF coding for the 2b protein, which is expressed via a subgenomic RNA known as sgRNA4A. RNA3 encodes the movement protein MP in a 5′ ORF and the coat protein CP in a 3′ ORF. CP is expressed via a subgenomic RNA known as sgRNA4.

Asparagus virus 2 (AV2) is a member of ilarvirus subgroup 2. AV2 is transmitted by seeds and pollen, but also mechanically by sap-contaminated tools during harvesting [24,25]. It was

shown that AV2 is capable of infecting by sap inoculation at least twenty-seven species of herbaceous plants in seven families, including *Chenopodium quinoa* and various tobacco species, which makes AV2 a good model for ilarvirus research [26]. Despite the fact that AV2-infected asparagus plants do not display visible symptoms, infection leads to a significant reduction in yield resulting in serious economic losses [24,25].

RNA interference (RNAi; also called RNA silencing) is a conserved mechanism of regulation of eukaryotic gene expression based on the sequence-specific degradation of RNA. In addition, RNAi serves as a major host defence mechanism against viruses in plants [27–31]. It is based on the recognition of viral dsRNA, which is formed either via internal self-complementarity and/or as a replication intermediate. Viral dsRNA is cleaved by host Dicer-like (DCL) proteins into 21–24 nt small interfering RNAs (siRNAs). siRNAs are subsequently incorporated into the host RNA induced silencing complex (RISC) and guide sequence-specific degradation and/or translational arrest of corresponding viral RNAs [31–33]. Cellular RNA-dependent RNA polymerases (RDRs), notably RDR6, can amplify virus-derived primary siRNAs to generate secondary siRNAs [34,35]. Secondary siRNAs are not only indispensable to further increase the effect of RNAi but also for its systemic spread [34,36]. Antiviral RNAi is not limited to the infection foci; secondary siRNAs move between cells via plasmodesmata and long distance via phloem to make non-infected tissues resistant to the virus spread or to a subsequent infection of the same virus [37–42].

To counteract RNAi, viruses have evolved viral suppressors of RNA silencing (VSRs). VSRs do not possess any universal structural similarity or common sequence motifs which suggests that they appeared in different groups of viruses as a result of convergent evolution. The mode of action of VSRs is also very diverse and they target various steps of the RNAi pathway [43–45]. The ability to suppress RNAi is often found as an additional activity of viral proteins with another established function and in most cases this function requires a close association with nucleic acids such as found in movement proteins (reviewed in Ref. [46]), coat proteins [47,48] or viral replicase proteins [49,50]. VSRs have been identified in almost all groups of plant viruses and they provide one of the most widespread viral counter-defence strategies [29].

AV2 is, like many ilarviruses, seed- and pollen-transmissible [22]. Moreover, it has been shown that AV2 is able to infect meristems [25]. Entry into meristems might serve as a gateway for vertical transmission though it is not imperative. The majority of plant viruses are excluded from meristems and are transmitted only horizontally. Factors limiting viral entry into meristems can be roughly divided into two groups: RNAi factors–with RDR6, salicylic acid and RDR1 as key players–and factors regulating the size and structure of plasmodesmata within meristems [51–53]. Viruses that infect meristems often encode weak VSRs, and it was suggested that incomplete suppression of RNAi allows meristematic entry of the virus [54–57]. Some VSRs regulate both vertical transmission and meristematic entry of the virus, suggesting that those processes are linked and RNAi-dependent [58–60]. AV2, like many vertically transmitted viruses, does not induce strong symptoms [61,62,25]. This, taken together with the ability of AV2 to enter meristems suggests that AV2 possibly suppresses RNAi in a mild and incomplete way. It has been shown that AV2 2b has a VSR activity and that it suppresses systemic RNAi [63]. Moreover, it was hypothesised that suppression of systemic RNAi is sufficient to ensure persistent infection and vertical transmission of AV2. Nevertheless, this does not exclude the possibility of another mechanism for the suppression of local RNAi during AV2 infection. Indeed, VSRs involved in vertical transmission and/or meristematic entry tend to be suppressors of local RNAi [54–60]. Some viruses encode more than one VSR that target different steps of the silencing pathway [64–68] or act cooperatively [69]. This strategy allows more sophisticated regulation of RNA silencing.

Using comparative genomic analysis, we show that many ilarviruses contain an additional ORF, here termed RT ("readthrough") immediately downstream of the CP stop codon, that we predicted to be expressed as a CP-RT fusion via readthrough of the CP stop codon. Using AV2 as a model ilarvirus, we show that CP-RT is produced upon virus infection, acts as a weak VSR, and we find that this function is indispensable for the establishment of persistent infection.

## Results

### Comparative genomic analysis suggests the presence of a CP readthrough domain in many ilarvirus groups

The three genome segments (RNA1, RNA2 and RNA3) of ilarviruses and the closely related alfamoviruses characteristically have shared terminal sequences, particularly in the 3′UTR. In AV2, for example, the terminal ~193 nucleotides are almost completely conserved between all 3 segments (SA Data). These sequence regions contain multiple AUGC motifs separated by RNA stem-loops that together bind to CP [70–72] (SA Data). The shared 3′UTR sequence normally begins shortly after, or even upstream of the stop codon of the 3′-most coding sequence on the segment, consistent with a model whereby RNA viruses are under strong selective pressure to delete any sequence that does not contain functional non-coding or coding elements.

During an analysis of ilarvirus sequences we noticed that, in a number of ilarviruses, there is a substantial insert in RNA3 between the CP stop codon and the shared 3′UTR sequence when compared to RNA1 and RNA2. Furthermore, for these RNA3 sequences, the next downstream CP-frame stop codon is positioned close to the start of the shared 3′UTR sequence (e.g. see Fig 1 and SA Data for AV2). Page 2 of SB Data shows an alignment of AV2 with closely related reference sequences, illustrating the insert between the CP stop codon and the start of the conserved 3′UTR elements. As for AV2, in all the sequences the next downstream CP-frame stop codon closely corresponds to the end of the insert region. We reasoned that an inserted region present across multiple divergent sequences must contain a functional element, either coding or non-coding. Given that the inserted region consistently corresponds to a CP-frame open reading frame, we hypothesised that it most likely represents a coding region. Given the absence of a conserved in-frame AUG codon, and the presence of a "leaky" CP stop codon (UGA C in all eight of these sequences and UGA CUA in four of them) we hypothesised that a proportion of ribosomes might read through the CP stop codon, terminating translation on the downstream in-frame stop codon, to produce a CP-RT protein.

We extended our analysis to all ilarvirus sequences available in the NCBI nr/nt and RefSeq databases as of 11 November 2022. We grouped ilarvirus sequences into clusters by applying BLASTCLUST [73] to CP amino acid sequences with an 80% identity threshold, as described in Materials and Methods. Additionally, clusters 5, 8 and 17 were each split into two subclusters (5a, 5b, etc.) due to the presence of two NCBI RefSeqs in each of the initial clusters. Cluster 6 was also split into two subclusters because sequences in this cluster appeared to divide into two groups–those with or without an insert in the RNA3 3′UTR. In each cluster we chose a reference sequence (see Materials and methods). SC Data presents (a) an intersegment alignment of the 3′UTR sequences of reference RNA 1, 2 and 3 sequences (where RNA1/RNA2 3′UTR sequences are available); and (b) for all available RNA3 sequences in the cluster with 3′UTR coverage (see Materials and methods), the context of the CP stop codon, and the amino acid translation of the ORF in-frame with and immediately 3′ of the CP stop codon whether or not readthrough is proposed to occur. The presence of a lengthy insert (or inserts) in RNA3 when the 3′-of-CP region was compared with the 3′UTRs of RNAs 2 and 3 was taken as a proxy for the putative presence of a RT domain in that cluster.

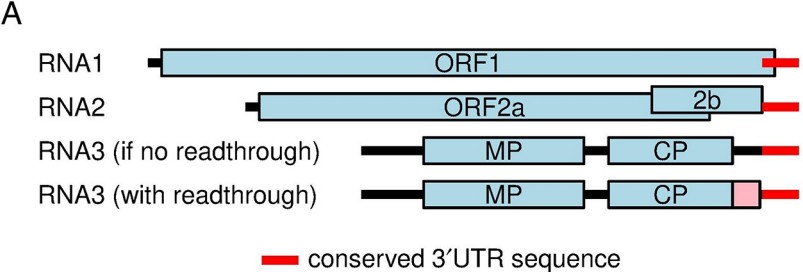

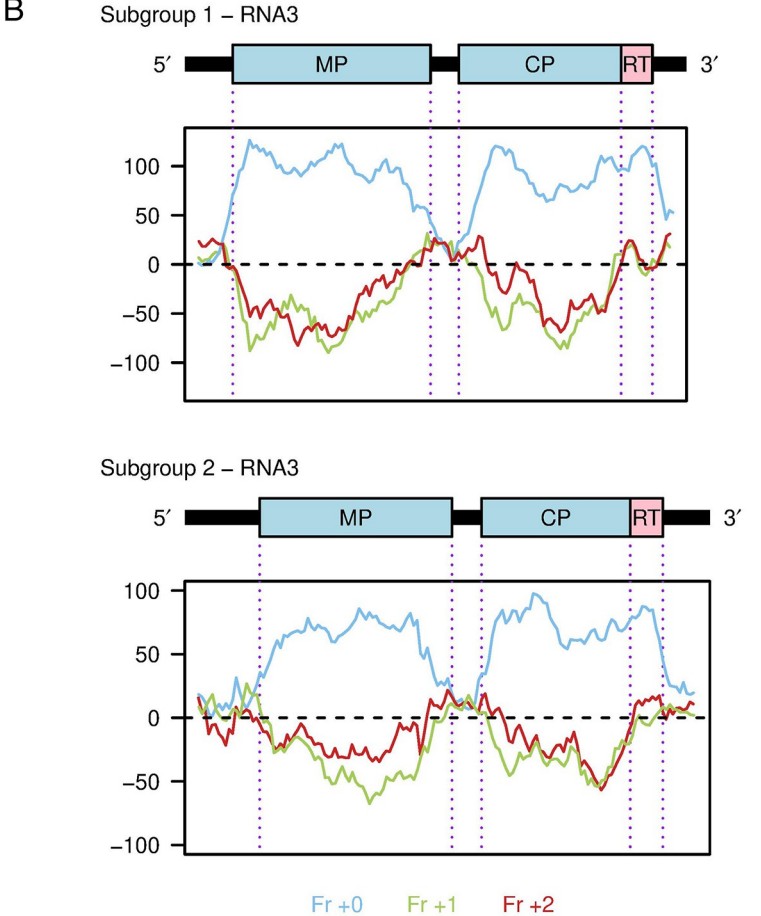

Fr +0      Fr +1      Fr +2

**Fig 1. A putative RT domain in ilarvirus RNA3. (A)** Map of the AV2 genome. Coding ORFs 1, 2a, 2b, MP (movement protein) and CP (coat protein) are annotated in light blue. The 193 nt 3′UTR sequence that is shared between all three segments is annotated in red. Note that in RNA1, this sequence overlaps with ORF1. In the old annotation, there is a substantial noncoding sequence (black) between the end of the CP ORF and the start of the shared 3′UTR sequence. The putative readthrough (RT) ORF (pink) fills this apparent gap. **(B)** Three-frame coding potential analysis of RNA3 as measured using MLOGD. Positive scores indicate that the sequence is likely to be coding in the given reading frame (blue–frame of the MP, CP and RT ORFs; green–+1 frame; red–+2 frame).

We found that some ilarviruses have a lengthy insert between the CP stop codon and the shared 3′UTR sequence whereas others did not. Where there was an insert sequence, there was always a lengthy potential in-frame readthrough extension of the CP ORF (SC Data) whereas, where there was no insert, the next CP-frame stop codon was often close to the CP stop codon (SA Table, SA Fig) or the corresponding ORF overlapped into the shared 3′UTR elements consistent with it being a spurious non-coding ORF (e.g. see cluster 2 in SC Data). Therefore, we hypothesised that those ilarvirus species with a lengthy insert in RNA3 (when the 3'-of-CP region is compared with the 3′UTRs of RNAs 2 and 3) have a RT domain, whereas those ilarviruses without such an insert lack a RT domain.

We further analysed the RT ORF for coding potential using the gene finding program MLOGD [74]. MLOGD analyses codon substitutions across a phylogenetic tree, and compares the likelihood of the observed substitutions under a null model (sequence is non-coding) and an alternative model (sequence is coding in a given reading frame), using the principle that synonymous substitutions and conservative substitutions (i.e. codons that code for amino acids with similar physicochemical properties) are more probable than non-conservative substitutions in typical coding sequences. Three-frame 'sliding window' analysis of RNA3 alignments for subgroup 1 or subgroup 2 reference sequences revealed a strongly positive coding signature in RT, similar to that observed in the MP and CP ORFs (blue lines, Fig 1B).

We found that viruses with a putative RT domain had CP stop codons that were typical of leaky stop codons, e.g. in the cluster reference sequences the CP stop codon contexts are UAG CAR YYA in clusters 5a, 5b and 6b; UGA CUA in clusters 1, 3, 8a, 8b, 13, 15, 16, 17a, 23, 27, 29 and 37; UGA CGG in cluster 10; and other, ostensibly less leaky UGA CNN sequences in clusters 4, 11, 14, 17b, 18, 30, 33, 34, 35 (SA Table). Only in clusters 12, 20, 25 and 36 (all UGA A) was the immediate stop codon context not typical of known leaky sites. In contrast, viruses without a putative RT domain had non-leaky stop codons, e.g. among the cluster reference sequences two have UAA, five have UAG not followed by CAR YYA, and three had UGA followed by C, G and U respectively (SA Table).

Fig 2 shows how the presence (red) or absence (black) of a putative RT domain distributes across the ilarvirus phylogeny based on a CP amino acid sequence alignment (see SC–SE Figs) for trees based on MP, ORF1 and ORF2a, respectively). In a couple of cases, an absence of appropriate RNA1/RNA2 sequence data meant that the presence or absence of an insert in RNA3 could not be determined (Fig 2 and SB–SC Figs, purple). Recombination and/or segment reassortment during ilarvirus evolution could lead to disparities among trees. Thus the CP tree likely most closely reflects the evolutionary history of the RT domain, although it is possible that recombination could occasionally occur between CP and RT. The latter possibility aside, the CP tree suggests that a RT domain has independently evolved on more than one occasion. Most of the RT-containing sequences can be grouped into one clade with a judicious re-rooting of the CP tree (red oval on Fig 2). However, the presence of a putative RT domain in apple mosaic virus (cluster 6b) and tea plant line pattern virus (cluster 37) represent– according to the CP tree–independent evolutionary events. Notably, cluster 6a (12 sequences) and cluster 6b (8 sequences) both represent apple mosaic virus. The latter have a 58 nt insert at the CP stop codon that contains the UAG CAR YYA Type I readthrough motif whereas, in the regions flanking the insert, cluster 6a and cluster 6b sequences align well with each other (see SC Data, cluster 6b). The resulting RT domain bears no similarity to the RT domains of other ilarviruses (SF Fig). Thus this RT domain does indeed appear to have evolved independently and more recently.

Many of the RT domain peptide sequences contain potential zinc finger motifs such as CxxC[. . .]CxxC (C = Cys; SF Fig) [75,76]. In many cases HHpred [77] predicted homology to zinc finger domains in known proteins whereas in other cases no significant homologies were

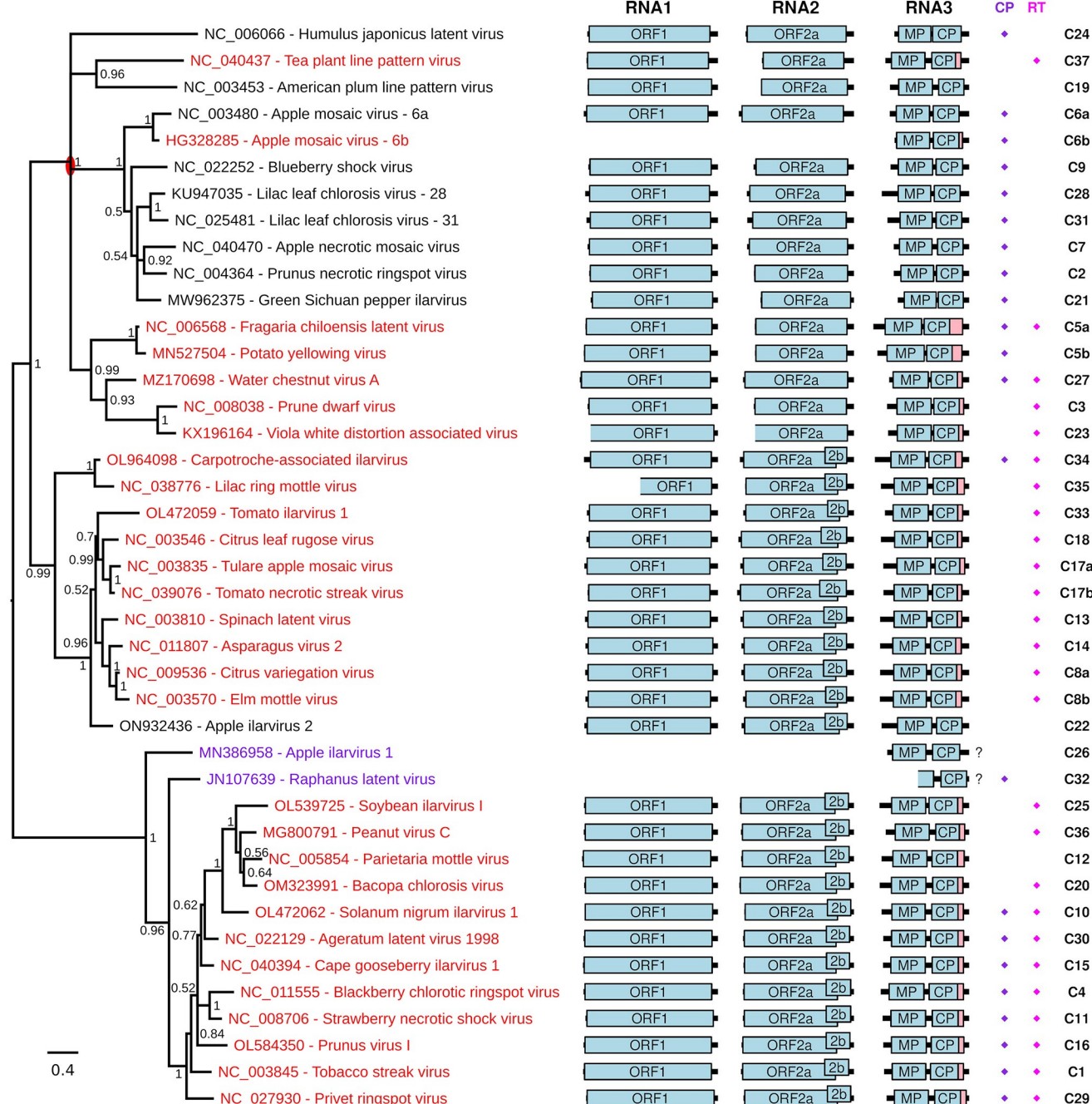

**Fig 2. Ilarvirus phylogenetic tree and genome maps.** At left is shown a phylogenetic tree of ilarvirus CP amino acid sequences–derived from one RNA3 reference sequence for each of 41 sequence clusters and subclusters (see SC Data). Reference sequences that have a putative RT domain (as defined by the apparent presence of a lengthy insert in RNA3 when the 3′-of-CP region is compared with the 3′UTRs of RNAs 2 and 3; see SC Data) are in red. Sequences for which this is unknown due to the absence of appropriate RNA1/RNA2 sequence data are in purple. Sequences that are proposed not to have a RT domain are in black. The tree is midpoint rooted and nodes are labelled with posterior probability values. The red oval indicates a possible re-rooting of the tree that groups together most of the sequences with a putative RT domain. Note that apple mosaic virus and lilac leaf chlorosis virus each occur twice on the tree since each species has representatives in >1 clusters (indicated by the cluster suffixes 6a/6b and 28/31, respectively; see SC Data for details). Genome maps are shown at right with ORFs 1, 2a, 2b, MP and CP indicated in pale blue and the putative RT domain, where present, in pink. "?"s indicate that the presence of a putative RT domain is unknown. Note that sequences EU919668, JN107639, KX196166 and KX196165 (at least) are not coding-complete; incomplete ORFs are indicated by the omission of the 5′ border of the corresponding ORF box. In addition, various sequences are missing various amounts of the 5′ and/or 3′UTR. At far right are the cluster numbers (C1, C2, etc). Diamonds indicate the presence of a putative zinc finger near the N-terminus of CP (purple) or within the RT domain (pink) as defined, simplistically, by the presence of a cluster of at least four Cys/His residues with at least two of them being Cys.

detected (SB Table). It is unclear whether these predicted zinc finger homologies represent horizontal transfer from host to virus or simply convergent evolution (i.e. not true homology) of a relatively simple motif. Zinc fingers come in a variety of forms, but are expected to contain (at least) four Cys and/or His residues that coordinate a zinc ion [75,76]. The Cys and His residues are often positioned as two pairs, with at most two of the four residues being His. While CxxC[. . .]CxxC is one common motif, there are many permissible variations in positioning and spacing of the Cys and His residues [75,76]. Thus, in cases where HHpred did not predict "homology" to known zinc finger domains, but there are nonetheless four or more Cys/His residues present, it is difficult to say definitively whether or not the sequence contains a zinc finger. However, three of the putative RT domains definitely lack zinc finger motifs, in the sense that they contain fewer than four Cys/His residues. These three are represented by reference sequences MN527504 (1 Cys, 1 His; cluster 5b), HG328285 (0 Cys, 0 His; cluster 6b) and NC_005854 (2 Cys; cluster 12). Of the remaining RT domains, all contain at least four Cys residues except for cluster 37 (NC_040437) where there are 2 His + 2 Cys residues. Notably, in the CP phylogeny (Fig 2), NC_040437 and HG328285 do not cluster with the main RT-containing clade, suggesting that their RT domains evolved independently and likely have different functions. In contrast, an ancestral zinc finger motif may have been lost in the MN527504 and NC_005854 lineages.

As noted in the Introduction, some but not all ilarviruses encode a 2b protein in an ORF that overlaps the 3′ region of ORF2a. To test whether presence of RT might correlate with presence or absence of ORF2b, we annotated ORF2b in all our RNA2 reference sequences based on the presence of a long ORF at this location (SG Fig; ORF2bs annotated in Fig 2 range from 522 to 690 nt in length). As can be seen in Fig 2, many ilarviruses have both ORF2b and a RT ORF, whereas some have only RT and some have neither. Furthermore, some but not all ilarviruses have a zinc finger motif near the N-terminus of CP [78,22]. Whereas some sequences clearly have a cluster of Cys and/or His residues near the N-terminus of CP, other sequences have few or no Cys or His residues in this region. Due to the difficulty in defining definitive motifs for zinc fingers, we annotated onto Fig 2 all sequences with at least two Cys and four or more Cys/His in the first alignment block of SD Data as containing a putative CP zinc finger. Similarly, we annotated onto Fig 2 all sequences with at least two Cys and four or more Cys/His in the RT domain as containing a putative RT zinc finger. Once again, some ilarviruses have both CP and RT putative zinc fingers, whereas some have only one or the other, and some have neither.

## CP-RT coding capacity is indispensable for persistent systemic infection

Complete sequences of the three genomic segments of AV2 (3431, 2916 and 2332 nt, respectively) were determined by RACE-PCR on infected *Nicotiana occidentalis* leaf material (DSMZ, Germany) (SE Data). A full length AV2 cDNA clone was assembled in the pDIVA backbone [79] which is a binary vector dedicated for agrobacterium-mediated infection. This clone consisted of three plasmids coding for the three viral genomic RNAs and a fourth plasmid coding for coat protein (CP) (Fig 3A). The fourth plasmid was included because, for ilarviruses and the closely related alfamoviruses, initiation of infection requires not only the viral genomic RNA but also the viral CP [80,81]. Work with the alfamovirus, alfalfa mosic virus, has shown that the mechanism underlying this dependency is (at least in part) that binding of CP to the 3′ end of the non-polyadenylated viral RNAs is required to stimulate their translation [82,83,21].

Five *Nicotiana benthamiana* plants were agroinfected with the AV2 cDNA clone (Fig 3B). At 21 dpi, upper non-inoculated leaves were analysed by western immunoblotting to detect

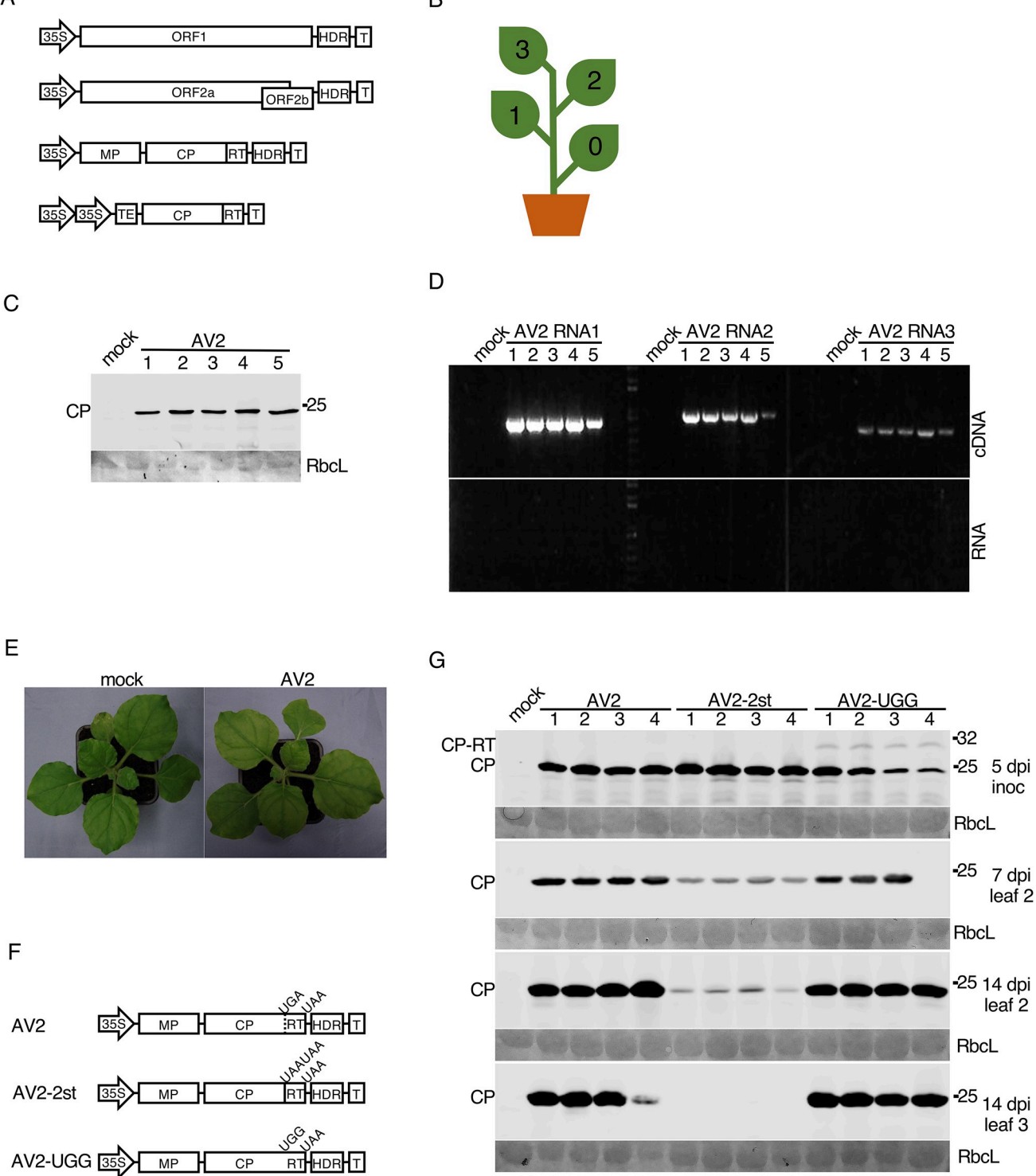

**Fig 3. Testing the infectivity of the AV2 full-length cDNA clone in *N. benthamiana* and mutagenesis of the RT domain. (A)** Schematic representation of the AV2 cDNA infectious clone. The full-length cDNAs of the AV2 genomic RNAs 1, 2 and 3 were inserted independently into pDIVA between the CaMV 35S promoter (35S) and a hepatitis delta ribozyme sequence (HDR) followed by a transcription terminator (T). The *cp-rt* coding sequence (including the CP stop codon) was inserted independently into pLH 7000 between double 35S promoters followed by the tobacco etch virus translational enhancer (TEV) at the 5′ end, and a transcription terminator (T) at the 3′ end. **(B)** Schematic representation of an infected *N. benthamiana* plant with indicated leaf positions where 0 indicates the inoculated leaf. **(C,D)** Detection of AV2 in the 3rd upper non-inoculated leaf at 21 dpi for five infected plants

by western blot against CP (panel C), and RT-PCR with primers for detection of RNAs 1, 2 and 3 (panel D). PCR on RNA without a reverse transcription step (panel E, bottom) served as a negative control. **(E)** Lack of visible symptoms on the upper non-inoculated leaves of a representative AV2-infected *N. benthamiana* plant (right) compared to a mock-inoculated plant (left). **(F)** Schematic representation of AV2 RNA3 mutants. **(G)** Detection of AV2 CP by western blot in plants infected with AV2, AV2-2st or AV2-UGG. Samples were collected from the inoculated leaf at 5 dpi, the 2nd non-inoculated leaf at 7 and 14 dpi, and the 3rd non-inoculated leaf at 14 dpi. Positions of CP and CP-RT are indicated on the left. In panels C and G, sizes of molecular weight markers are indicated on the right, and Ponceau red staining (lower panels) was used as a loading control.

AV2 CP (Fig 3C) and by RT-PCR to detect genomic RNAs (Fig 3D). Although inoculated plants displayed no visible symptoms at 21 dpi (Fig 3E), viral RNAs and CP were detected in every infected plant confirming the infectivity of the AV2 cDNA clone in *N. benthamiana*. Plants were monitored up to 47 dpi; they remained asymptomatic, but viral CP was detected in the upper leaves of all infected plants by western blotting (SM Fig). Thus we can conclude that AV2 infection in *N. benthamiana* is persistent.

To examine the role of the putative CP-RT protein, two mutant variants of the RNA3 clone were engineered (Fig 3F). In AV2-2st, the CP weak stop codon UGA was substituted with tandem strong stop codons (UAA UAA) to abolish readthrough. In AV2-UGG, the stop codon of CP was substituted with a tryptophan codon so that only CP-RT but not CP could be produced. Mutant variants of the RNA3 were agroinfiltrated into *N. benthamiana* along with the RNA2, RNA1 and CP plasmids. Western blotting against CP revealed that in AV2-UGG inoculated leaves, where virus spread and replication could be supported by the CP provided *in trans*, CP-RT was detected but at a relatively low level which could possibly be due to protein instability. In upper non-inoculated leaves, AV2-UGG did not show any difference in CP accumulation compared with the wild type AV2 and CP-RT was not detected (Fig 3G). This suggests instability of the UGG mutation, as this mutation should not allow production of CP but only CP-RT once the virus has spread to non-inoculated leaves (the CP-expressing plasmid, present in the inoculated leaves, is not expected to spread as it lacks the viral native 3′UTR sequence). Sequencing of the virus progeny indeed revealed 100% reversion (*n* = 4 plants) of the UGG mutation in AV2-UGG to the wild type UGA. We observed that AV2-2st-infected plants had a reduced CP accumulation in the 2nd non-inoculated leaf compared to plants infected with wild type AV2 at 7 and 14 dpi and that CP was undetectable in the 3rd non-inoculated leaf at 14 dpi (Fig 3G). Sequencing of virus progeny from the 2nd leaf at 14 dpi confirmed full stability of this mutation. Thus, our results show that mutation preventing CP-RT production decreases virus accumulation and prevents the establishment of a persistent systemic infection.

To further test the stability of the wild type AV2 and AV2-2st viruses, we performed RNA-seq. *N. benthamiana* plants were agroinfected with AV2 or AV2-2st and systemically infected leaves were harvested at 7 dpi. We prepared RNA-seq libraries for three separate plants for each virus. Reads were mapped to the AV2 or AV2-2st genomes. In all six libraries, the last nucleotide of RNA3 appeared to be lost, as evidenced by the density of RNA-seq mapping to the last nucleotide relative to the second last nucleotide being 0.015–0.037 in RNA3 but 0.425–0.794 and 0.412–0.832 in RNAs 1 and 2 respectively. In our infectious clone, RNAs 1 and 2 end in GGAGAUGC-3′ whereas RNA3 ends in GGAGAUGCU-3′, i.e. RNA3 has an extra 3′-terminal nucleotide compared to RNAs 1 and 2. The fact that this is lost during infection suggests that it is a spurious addition and not generally present in the natural virus. Relative to the sequences of the infectious clone (GenBank accessions OR820775, OR820776 and OR820777), across all six libraries only six other SNPs were identified with frequencies above 10% in a given library, and all of these had frequencies below 25%. None of the six SNPs occurred at the sites mutated to generate the AV2-2st mutant (i.e. there was no evidence for reversion or

pseudoreversion in this experiment). To test for possible insertions/deletions that might be missed by bowtie mapping to a reference sequence, we also *de novo* assembled the reads with the SPAdes genome assembler [84] in rnaviralSPAdes mode [85]. Except for the AV2 replicate 3 RNAs 1 and 2 where contigs were fragmented due to poorer coverage (SH Fig), single long contigs were assembled for each of RNAs 1, 2 and 3 in each library, with each assembly full length except for up to 45 and up to 161 nt of missing 5′- and 3′-terminal sequence, respectively. In these assemblies there were no indels relative to the respective AV2 and AV2-2st genomes.

The density of RNA-seq reads mapping to the viral genomic RNA is shown in SH Fig. We observed average RNA1:RNA2:RNA3 positive-sense density ratios of 1.1:1:3.9 and 1.9:1:5.3 across the three wild type and three AV2-2st samples, respectively (SH(ii) Fig). Very little negative-sense RNA was detected (0.013–0.036% of total vRNA) and, while excess density could be observed in the sgRNA region of RNA2 (i.e. ORF2b; sgRNA4A), this excess density was often weak and any excess density in the sgRNA region of RNA3 (i.e. CP; sgRNA4) was not obvious. Viral RNA in leaves may comprise a mixture of translatable mRNA, positive-sense and negative-sense RNA in replication complexes, and RNA packaged in virions (plus, potentially, sequestered and/or degraded RNA; the range in read lengths for these samples should however be large enough to exclude products of RNAi-mediated degradation). Virion-packaged RNA likely comprises mainly full-length positive-sense genomic RNAs 1, 2 and 3; however sgRNAs 4 and 4A are also known to be packaged at varying levels (potentially depending on species and other factors) [81,21]. RNAs 1, 2 and 3 –which are packaged within separate particles–are also not packaged at equimolar frequencies. For example, for the subgroup 2 ilarvirus, spinach latent virus, Xin et al. detected substantially more packaged RNA3 than RNAs 1 and 2 (Fig 3 of Ref. [86]). As for sgRNAs 4 and 4A, Xin et al. found sgRNA4 to be more efficiently packaged than sgRNA4A whereas sgRNA4A was much more abundant than sgRNA4 in leaves.

## CP-RT is expressed *in vitro* in wheatgerm extract and *in planta* upon AV2 infection

To determine whether the bioinformatically predicted readthrough event genuinely occurs, we performed *in vitro* translation of sgRNA4. 5′RACE-PCR indicated that the 5′ end of the sgRNA4 might lie 53 nt upstream of the CP AUG, resulting in 5′-GUCUUG which partly resembles the 5′ ends of the genomic RNAs (5′-GUAUUG in RNAs 1 and 2, GUAUUC in RNA3) (SB Data, SE Data). The sgRNA4 cDNA as well as cDNAs with the UGG and 2st mutations (sgRNA4-UGG and sgRNA4-2st) were cloned under the T7 promoter (Fig 4A). *In vitro* translation of sgRNA4 in wheatgerm extract led to the accumulation of CP and CP-RT, though the amount of CP-RT was considerably lower than that of CP. Translation of sgRNA4-UGG and sgRNA4-2st resulted in the accumulation of only CP-RT and only CP, respectively (Fig 4B). Thus, *in vitro* readthrough on wild type sgRNA4 occurs but is extremely inefficient.

Considering the difference of accumulation of CP and CP-RT in the *in vitro* translation system (Fig 4B), it is not surprising that we could not detect CP-RT in AV2-infected plants using polyclonal antibodies raised against virus particles (e.g. Fig 3G). Therefore, to detect CP-RT upon infection we made an AV2-myc clone in which a myc tag was added on the C-terminal end of CP-RT (Fig 4C). Using anti-myc antibodies, CP-RT-myc was detected in the 2nd non-inoculated leaf at 4 and 7 dpi but not in the 3rd non-inoculated leaf at 14 dpi in AV2-myc infected plants. CP-RT-myc was detected as two closely migrating bands which might reflect protein modifications or partial degradation. The myc tag prevented the establishment of persistent infection. Western blotting for CP revealed that CP had the same pattern of

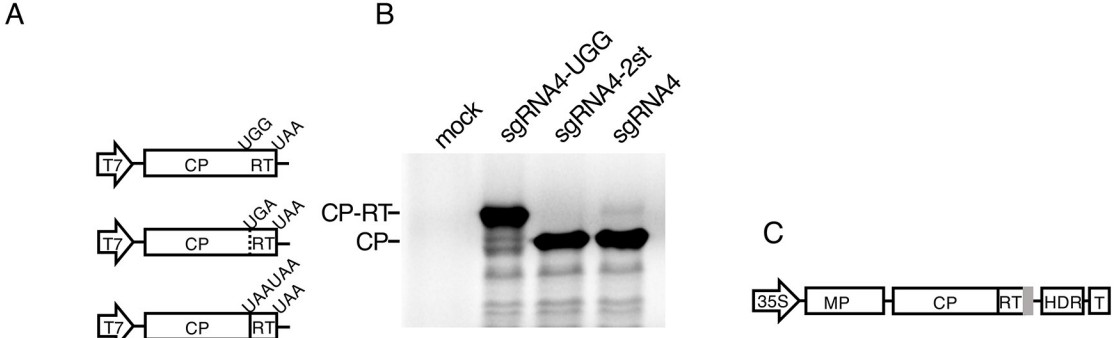

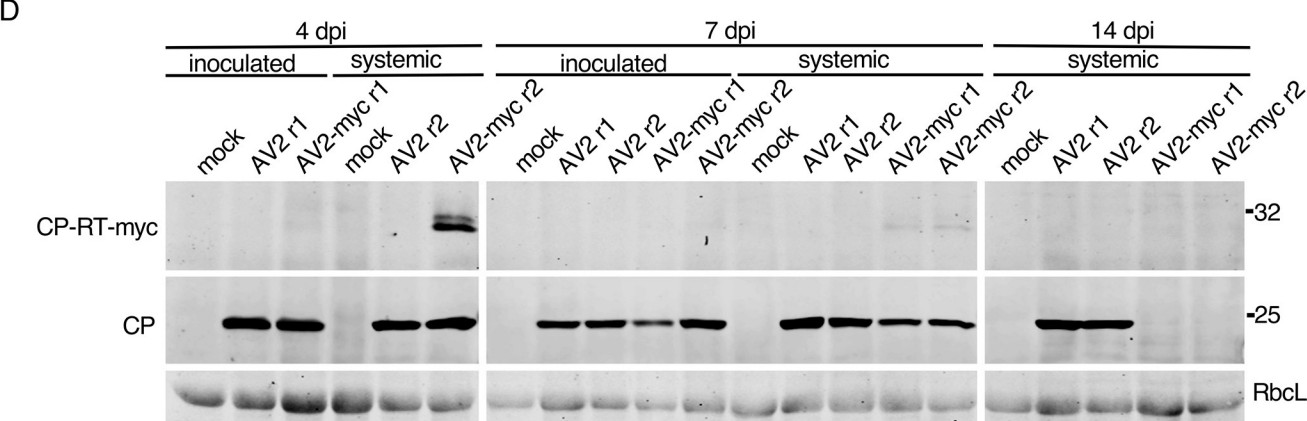

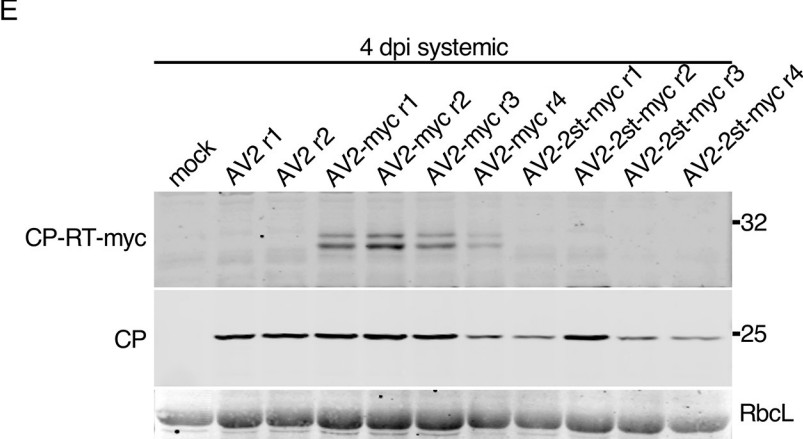

**Fig 4. Detection of CP-RT *in vitro* and *in vivo*. (A)** Schematic representation of sgRNA4 and its UGG and 2st mutant clones under a T7 promoter. **(B)** SDS-PAGE of proteins translated in wheatgerm extracts from *in vitro* transcripts of sgRNA4 and its two mutants. Mock–no RNA added. After drying the gel, proteins radioactively labelled with [$^{35}$S]Met were detected using a phosphorimager. **(C)** Schematic representation of the AV2 RNA3 with a tag (grey rectangle) appended to the 3′ end of the CP-RT gene. **(D)** Detection of CP and CP-RT-myc by western immunoblotting in plants infected with AV2 or AV2-myc. Samples were collected from the inoculated leaf and the 2nd upper non-inoculated leaf at 4 and 7 dpi, and the 3rd upper non-inoculated leaf (see Fig 3B) at 14 dpi. **(E)** Detection of CP and CP-RT-myc by western blotting in extracts of plants infected with AV2, AV2-myc or AV2-2st-myc. Samples were collected from the 2nd upper non-inoculated leaf at 4 dpi (see Fig 3B). Positions of CP and CP-RT-myc are indicated on the left. Sizes of molecular weight markers are indicated on the right in panels B, D and E. Ponceau red staining of the large Rubisco subunit (RbcL) was used as a loading control in panels D and E.

accumulation as CP-RT-myc (Fig 4D). Hence it is possible to detect CP-RT *in planta* even though the addition of the tag prevented establishment of persistent infection and decreased the level of virus accumulation (Fig 4D). As an additional control, the myc tag was introduced in AV2-2st to produce the AV2-2st-myc virus. As expected, no proteins of CP-RT size were detected with anti-myc antibodies. AV2-2st-myc presence was confirmed by western blot against CP (Fig 4E).

In case the myc tag specifically prevents the normal functioning of CP-RT, we also made AV2 and AV2-2st clones with HA tags. We obtained the same results with HA tags as with myc tags (SI Fig). Therefore, we conclude that CP-RT is expressed upon AV2 infection and the addition of a tag on its C terminus results in a similar phenotype as for knockout of CP-RT expression.

### Ribosome profiling further supports readthrough of the CP stop codon

To provide additional support for translation of the RT ORF during virus infection, but without the addition of tags, we agroinfected *N. benthamiana* plants with AV2 or AV2-2st and performed ribosome profiling of systemically infected leaves at 12 dpi. Ribosome profiling (Ribo-seq) is a high-throughput sequencing technique that globally maps the footprints (ribosome protected fragments, or RPFs) of initiating or elongating 80S ribosomes on mRNAs [87,88].

An analysis of read length distributions (Fig 5A) showed a strong peak in read lengths at 27 and 28 nt, but also a substantial number of reads at both shorter and longer lengths. The longer and shorter reads may represent poorly trimmed and over trimmed RPFs, respectively, but more likely a substantial fraction of these reads derive from non-RPF contamination such as footprints of other ribonucleoprotein complexes. Longer reads were more abundant among the virus-mapping reads than the host-mapping reads. To increase the proportion of true RPFs to contamination we therefore selected only 27 and 28 nt reads for further analysis. It should be noted that we still expect some contamination among such reads. A characteristic of true RPFs is that, when mapped to coding sequences, their distribution reflects the triplet periodicity of genetic decoding. In good quality preparations a high proportion of RPF 5′ ends for a specific read length size class will map to a specific position (i.e. 1st, 2nd or 3rd nucleotide) of codons. For plant ribosome profiling, the 5′ ends of 28 nt reads typically map to the first position of codons whereas the 5′ ends of 27 nt reads typically map to the second position of codons [89,90]. We observed similar phasing patterns for our samples (Fig 5B, SJ(i) Fig), with a mean of 88% of reads mapping to the preferred phase for host-mapping reads. This was reduced to a mean of 72% for virus-mapping reads, again indicating a modest level of non-RPF contamination among the virus-mapping reads.

Another way to gauge ribosome profiling quality is to assess the density of reads in 3′UTRs since, even for poorly phased samples, an absence of reads in 3′UTRs is an indicator that reads represent *bona fide* RPFs. For host-mapping reads, the distribution of read 5′ ends relative to initiation and termination codons was summed over all host mRNAs to produce average mRNA read density profiles (Fig 5C, SJ(ii) Fig). The density of reads within host 3′UTRs was very low for all samples, again supporting the high quality of these samples when restricted to 27 and 28 nt host-mapping reads. As observed by Chung et al. [89] for *Chlamydomonas* and Hsu et al. [90] for *Arabidopsis*, we find that plant ribosomes at the stop codon typically have RPFs that map to a different phase from elongating ribosomes of the same read length–the 2nd nt position for 28 nt reads and the 3rd nt position for 27 nt reads.

For 27 and 28 nt reads combined, we observed 3.9 and 4.3 million reads mapping to host mRNA in AV2 and AV2-2st samples, respectively, and 60843 and 5651 reads mapping to virus positive-sense RNA (SC Table). Within ORFs 1, 2a, 2b, MP and CP, 27 and 28 nt reads

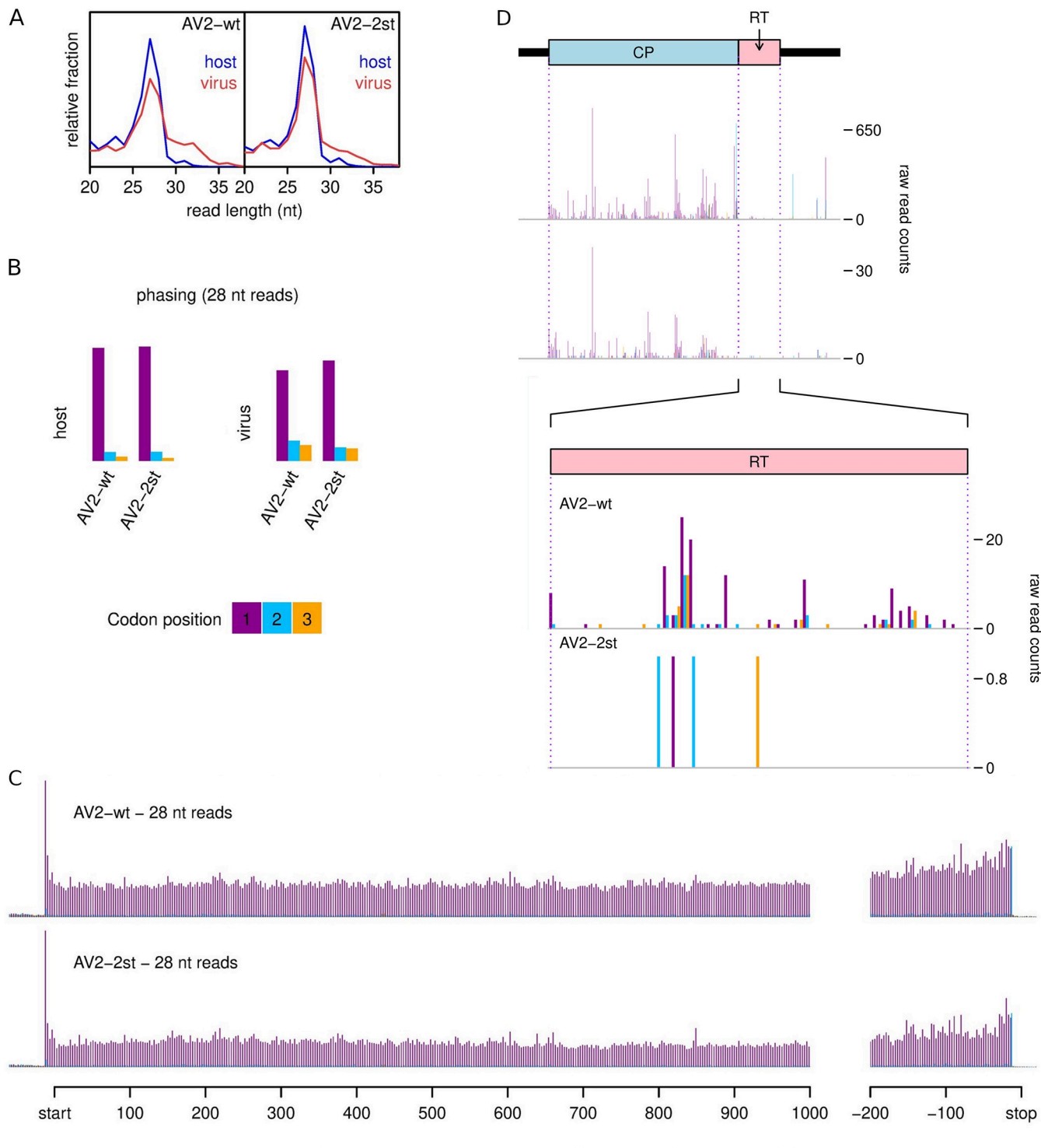

**Fig 5. Ribosome profiling of systemically infected leaf tissues from *N. benthamiana* plants agroinfected with AV2 or AV2-2st.** **(A)** Relative length distributions for Ribo-Seq reads mapping to virus (red) and host (blue) mRNA coding regions. **(B)** Phasing of 5′ ends of 28 nt reads that map to the viral ORFs (excluding dual coding regions) or host mRNA coding regions. **(C)** Histograms of approximate P-site positions of 28 nt reads relative to annotated initiation and termination sites summed over all host mRNAs. For panels C and D, see SJ Fig for the 27 nt read data. **(D)** Distribution of 28 nt reads on the 3′ half of RNA3 for AV2 and AV2-2st. Histograms show the positions of the 5′ ends of reads, with a +12 nt offset to map approximate P-site positions. Colours purple, blue and orange indicate the three different phases relative to the reading frame of the CP ORF. Therefore, consistent with panel B, most 28 nt reads map to the

purple phase in the CP ORF, and true ribosome protected fragments (RPFs) are expected to map to the purple phase in the RT ORF. Note that nucleotide-to-nucleotide variation in RPF counts may be influenced by technical biases besides ribosome codon dwell-times. Reads in the 3′UTR and a fraction of reads throughout the genome undoubtably derive from non-RPF contamination, and contamination is likely to be relatively more pronounced in the lowly expressed AV2-st (cf. SC Table). See SK Fig and SL Fig for full-genome plots and for 27 nt read data.

mapped predominantly to the expected phases (Fig 5D, SK, SL Figs). There were substantial occasional peaks in other phases, which may represent the effects of local ligation biases, local phasing biases (e.g. one fewer or one more nucleotide protected by the ribosome due to local sequence features) or local contamination hotspots. The sporadic nature of the density peaks also likely reflects ribosome pausing and ligation bias effects, besides shot noise, and is typical when viewing the density profile for any individual transcript [87,88,90]. Unusually, there was a substantial increase in density towards the 5′ ends of ORFs (most obvious in ORFs 1 and 2a) in the AV2-2st sample that was not apparent in the wild type AV2 sample (SK, SL Figs). One possibility is that this is simply an artefact accidentally introduced during sample preparation although, arguing against this, we did not observe a similar 5′ pile-up for host mRNA (SJ(ii) Fig). Other possible explanations may be that AV2-2st is translationally defective or that AV2-2st viral RNAs are being cleaved/degraded during the course of being translated so that fewer ribosomes are able to reach the 3′ regions of transcripts. Footprints in the lengthy 5′UTR of RNA3 may represent ribosomes translating short uORFs (as seen in many Ribo-seq experiments) [91,92], and/or non-RPF contamination, and again were more apparent in the AV2-2st sample than the wild type AV2 sample.

Having assessed Ribo-seq quality, we looked at the data for evidence of RT ORF translation. The very low level of RT expression suggested from *in vitro* translation (Fig 4B) and our inability to detect CP-RT in wild type AV2 infection with anti-CP immunoblotting (Fig 3G), together with the presence of contaminating non-RPF reads in our virus-mapping data (see above) meant that caution was required. For AV2, we saw density within the RT ORF that included many distinct positions and these positions were predominantly in the expected phase (2nd nt positions of codons for 27 nt reads, 1st nt positions of codons for 28 nt reads; Fig 5D, lower panels of SL and SK Figs). In contrast, for AV2-2st, reads mapped to only a few distinct positions within the RT ORF (six for 27 nt reads, four for 28 nt reads) and the positions were not predominantly in the expected phase for RT ORF translation. Thus we concluded that the RT ORF is indeed translated during AV2 infection but not during AV2-2st infection.

## The RT domain of tobacco streak virus partially complements the AV2 RT domain

We have shown above that knockout of CP-RT or the addition of a tag at its C terminus make AV2 infection transient instead of persistent. Whereas AV2 is a subgroup 2 ilarvirus, tobacco streak virus (TSV) is in subgroup 1 and, although it was also predicted to have a RT domain, the RT sequence is quite different from that of AV2 (Fig 6A). To test whether the TSV RT domain can complement that of AV2, a chimera AV2-TSV_RT was created in which the RT domain of AV2 was substituted by the RT domain of TSV (SF Data). We agroinfected *N. benthamiana* plants with AV2-TSV_RT or AV2. AV2-TSV_RT was able to establish persistent systemic infection, with CP detected in upper non-inoculated leaves of four out of five infected plants even at 17 dpi (Fig 6B). Of note, the level of CP accumulation was slightly lower in the plants infected with AV2-TSV_RT compared to plants infected with AV2 at all time points

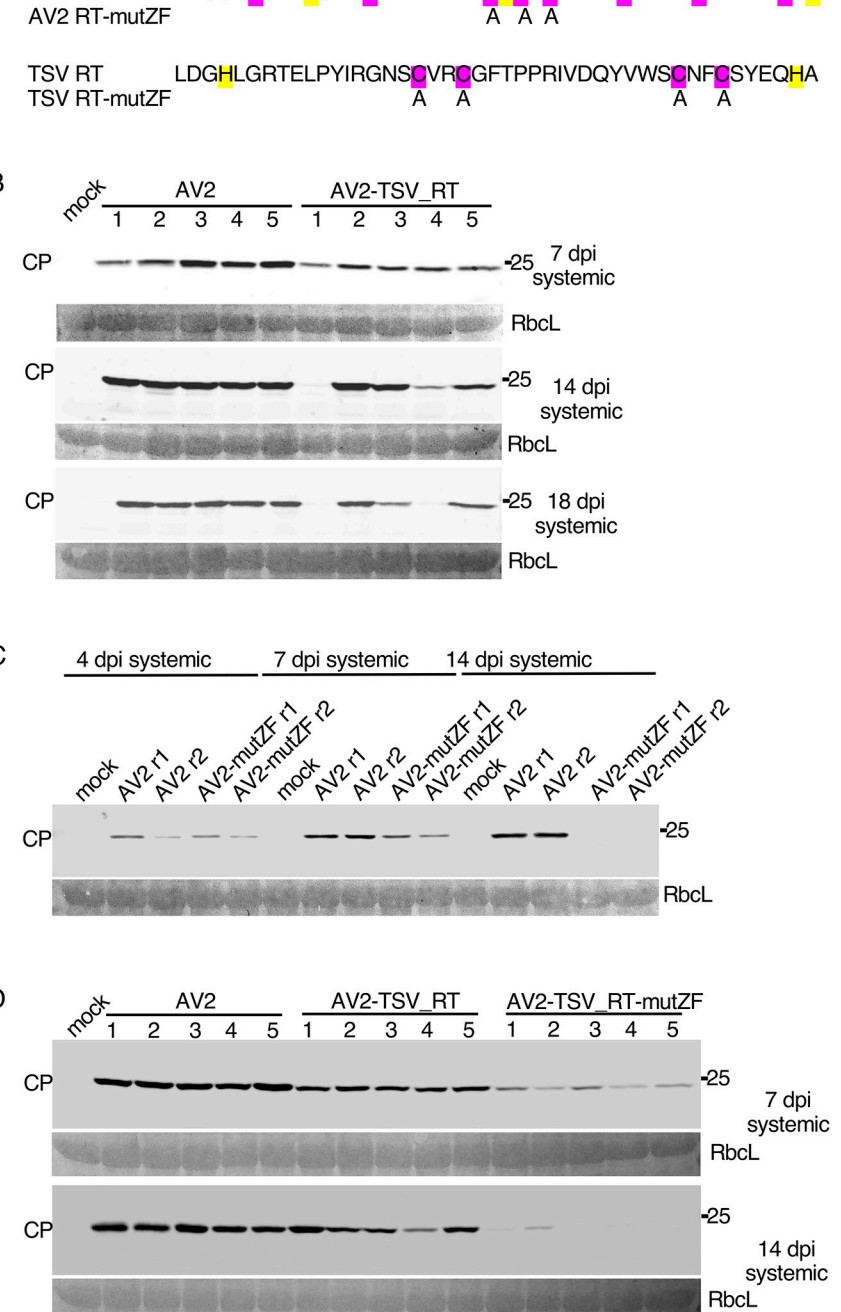

**Fig 6. Complementation of the AV2 RT domain by the TSV RT domain and mutagenesis of the putative zinc finger.** (**A**) Mutations introduced into the AV2 and TSV RT domains. Cysteines and histidines–potentially involved in zinc finger formation–are highlighted in purple and yellow, respectively. Replaced amino acid residues are indicated. (**B**) Detection of CP by western blotting in protein extracted from plants infected with AV2 or AV2-TSV_RT. Samples were collected at 4 and 7 dpi from the 2nd upper non-inoculated leaf, and at 14 and 18 dpi from the 3rd upper non-inoculated leaf. (**C**) Detection of CP by western blotting in extracts of plants infected with AV2 or AV2-mutZF.

Samples were collected at 4 and 7 dpi from the 2nd upper non-inoculated leaf, and at 14 dpi from the 3rd upper non-inoculated leaf. **(D)** Detection of CP by western blotting in protein extracted from plants infected with AV2, AV2-TSV_RT or AV2-TSV_RT-mutZF. Samples were collected at 7 dpi from the 2nd upper non-inoculated leaf, and at 14 and 17 dpi from the 3rd upper non-inoculated leaf. In panels B, C and D, sizes of molecular weight markers are indicated on the right, and Ponceau red staining of the large Rubisco subunit (RbcL) was used as a loading control.

assessed (Fig 6B). Viral RNA in systemically infected leaves was sequenced and the chimera was found to be stable. Thus the TSV RT domain partially complements that of AV2.

## A zinc finger motif within the RT domain is important for AV2 persistent systemic infection

Both AV2 and TSV RT domains contain a putative zinc finger. To test the functional significance of this motif we substituted cysteines by alanines in AV2 and AV2-TSV_RT, giving AV2-mutZF and AV2-TSV_RT-mutZF respectively (Fig 6A). First, we infected *N. benthamiana* plants with AV2-mutZF or AV2 and analysed virus accumulation in upper non-inoculated leaves by anti-CP western blot at three different time points (Fig 6C). Similar experiments were then performed with AV2-TSV_RT-mutZF alongside AV2 and AV2-TSV_RT (Fig 6D). Both zinc finger mutants AV2-mutZF and AV2-TSV_RT-mutZF had the same pattern as AV2-2st of transient virus accumulation in upper non-inoculated leaves (Fig 6C and 6D). This suggests that the zinc finger motif is an important functional element of CP-RT.

## CP-RT suppresses transgene-induced RNAi in transient silencing suppression assays

All mutations affecting the RT domain made AV2 infection transient instead of persistent: no CP-RT mutant virus was detected in the 3rd non-inoculated leaf two weeks after inoculation, whereas wild type AV2 could be detected in upper non-inoculated leaves for as long as 8 weeks after inoculation (SM Fig). Since RNAi is a key plant antiviral defence system, we hypothesised that AV2 persistence may be based on a capacity of CP-RT to suppress RNAi.

To test this hypothesis, we first performed a transient silencing suppression assay using an *Agrobacterium* co-infiltration method [93] to test whether CP-RT can suppress RNAi triggered by overexpression of *gfp-c3* [94] in *N. benthamiana* leaves. This reporter construct was co-agroinfiltrated with plasmids expressing VSR candidates or an empty plasmid into *N. benthamiana* leaves (Fig 7A). Sequences *cp-rt* (expressing CP and small amounts of CP-RT), *cp-rt-ugg* (expressing CP-RT only), *cp* (expressing CP) and *cp-rt-ugg-mutzf* (expressing CP-RT with mutated zinc finger) were assessed as VSR candidates. The AV2 2b protein has been shown to suppress systemic RNAi but not local RNAi [63], so we also included a plasmid expressing AV2 2b to confirm that it does not affect local RNAi. The known VSR p19 from tomato bushy stunt virus [95] was used as a positive control. GFP accumulation was assessed by evaluation of infiltrated leaves under UV light (Fig 7B) and *gfp* mRNA accumulation was measured by RT-qPCR at 4 dpi (Fig 7C). Co-expression with *cp-rt-ugg* led to a 5–10 fold higher level of GFP mRNA compared to co-agroinfiltration of *gfp* with an empty plasmid. Accumulation of *gfp* mRNA was not affected by co-expression with *cp*, *cp-rt-ugg-mutzf* or 2b and was drastically increased by co-expression with *p19* (Fig 7C). To further test our supposition that stabilisation of *gfp* mRNA by CP-RT is due to the suppression of RNAi, we analysed the accumulation of *gfp*-specific siRNAs by northern blotting. Unexpectedly, whereas co-agroinfiltration with *p19* almost abolished the accumulation of *gfp* siRNAs, we were unable to see a clear

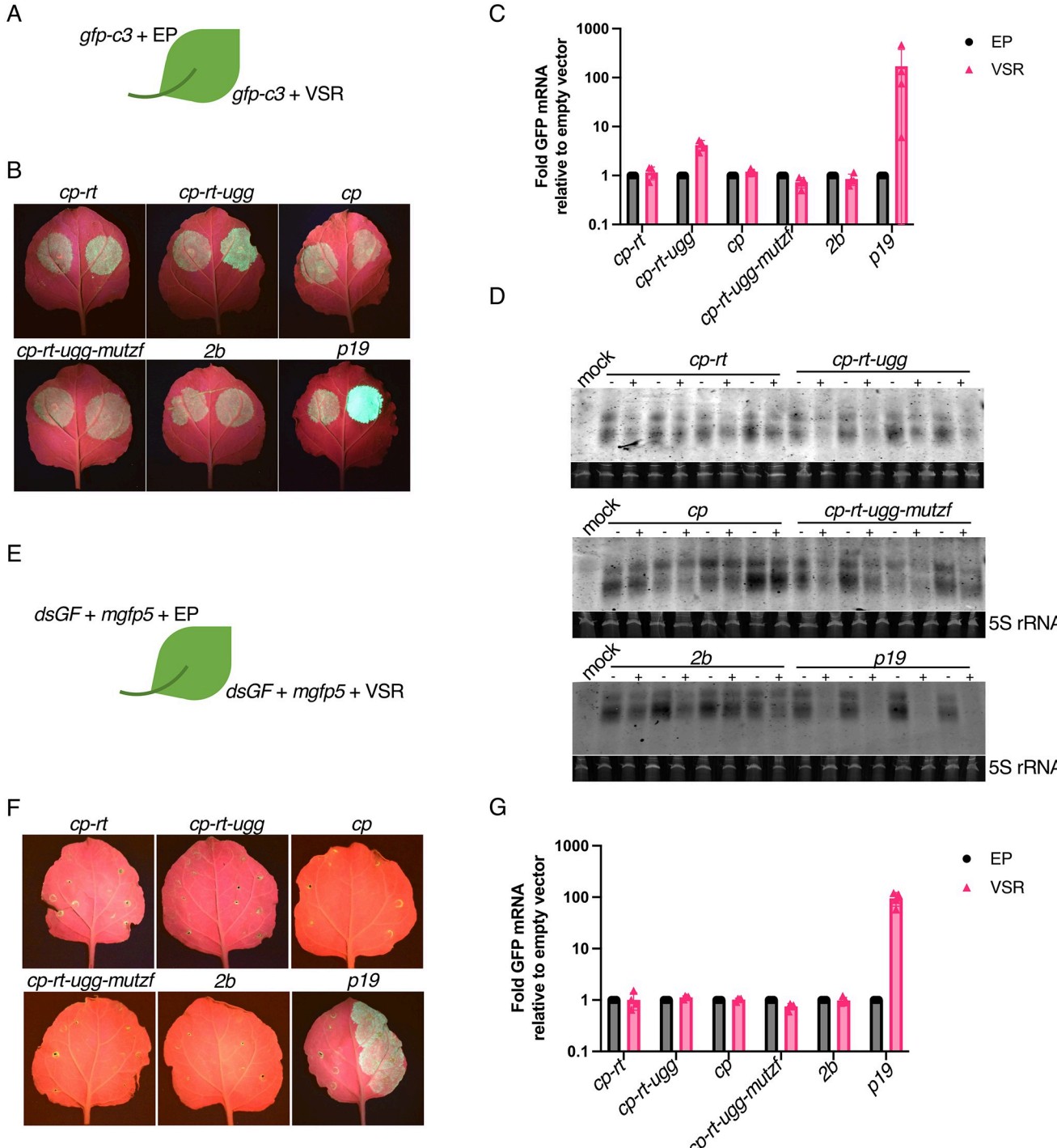

**Fig 7. Silencing suppression activity of *cp-rt*, *cp*, *cp-rt-ugg* and *cp-rt-ugg-mutzf* in an agroinfiltration assay. (A)** *gfp-c3* was introduced by agroinfiltration into *N. benthamiana* leaves along with a plasmid encoding a candidate VSR (right half of leaf) or the empty plasmid (EP) (left half of leaf). **(B)** Agroinfiltrated *N. benthamiana* leaves were imaged under UV light at 4 dpi. The agroinfiltrated constructs are indicated above the images. **(C)** RT-qPCR analysis of *gfp* mRNA accumulation. *Nb ACT-b* (GI:380505031) was used as a reference gene. Values represent the mean +/− SD (*n* = 4). **(D)** Northern blot analyses depicting the accumulation of *gfp*-specific siRNAs. A hydrolysed Cy5 labelled *in vitro* transcribed RNA fragment complementary to *gfp* was used as a probe. 5S ribosomal RNA was used as a loading control. **(E–G)** As for panels A–C except *mgfp5* and *dsGF* were used instead of *gfp-c3*.

unambiguous effect of *cp-rt-ugg*, *cp-rt*, *cp-rt-ugg-mutzf*, *cp* or *2b* on *gfp* siRNA accumulation (Fig 7D).

In the above-described experimental setup, RNA silencing was induced by overexpression of a heterologous mRNA. We used an additional assay to test for suppression of a different type of inducer of RNAi. In this assay, we used a dsRNA silencing inducer, *dsGF*, which consists of two copies of the 5′-proximal 300 nt of *mgfp5* in the sense and antisense orientations separated by an 8 nt spacer [93,96]. We transiently co-expressed in *N. benthamiana dsGF* with *mgfp5* and each VSR candidate or an empty plasmid (Fig 7E). Long dsRNA is a strong trigger of RNAi, and our observations of GFP fluorescence and *mgfp5* mRNA accumulation showed that only *p19* was able to suppress RNAi in this system (Fig 7F and 7G).

In conclusion, CP-RT expressed from the *cp-rt-ugg* plasmid suppresses RNAi induced by overexpression of single strand but not double strand heterologous RNA. The negative results obtained with *cp-rt-mutzf* show that the zinc finger motif is required for the VSR activity of CP-RT.

### RDR6 deficiency compensates for a lack of CP-RT

In *Arabidopsis thaliana*, RDRs–notably RDR1 and RDR6 –are implicated in intercellular and systemic spread of RNAi [97–99,41]. In *A. thaliana*, RDR1 and RDR6 function synergistically in antiviral defence [100]. *N. benthamiana* possesses a non-functional RDR1 gene [101] and the most important antiviral RDR is NbRDR6. Plants with reduced expression of NbRDR6 are more susceptible to a subset of viruses [41]. NbRDR6 plays an important role in preventing or limiting virus infection in shoot apical meristem, which suggests a role in plant recovery [102,103]. We have shown above that CP-RT expressed alone is able to suppress RNA silencing in a system where the inducer is a highly abundant single-stranded RNA (Fig 7). This type of RNA silencing particularly relies on activity of the host RDRs [103,104]. Together with the observation that all mutations affecting CP-RT lead to a recovery-like phenotype of AV2 infection, this suggests that CP-RT interferes with an RDR6-dependent branch of the RNAi pathway. To test this hypothesis, we infected wild type and RDR6 knockdown (RDR6i) [105] *N. benthamiana* with AV2, AV2-2st or AV2-mutZF. In wild type *N. benthamiana* infected with AV2-2st or AV2-mutZF, CP was detected only at 7 dpi and completely vanished by 14 dpi (Fig 8A). In contrast, in RDR6i *N. benthamiana* infected with AV2-2st or AV2-mutZF, CP was detected at 14 dpi and even 21 dpi although at a lower level compared with AV2-infected plants (Fig 8A). The dynamics of virus accumulation in infected plants was also analysed by RT-qPCR with RNA3-specific primers (Fig 8B) and the result was consistent with western blot against CP. In accordance with our hypothesis that CP-RT is involved in suppression of RDR6-dependent RNAi and that the zinc finger motif is involved in silencing suppression by CP-RT, AV2-2st and AV2-mutZF were able to establish persistent systemic infection in RDR6i but not wild type *N. benthamiana*.

To test whether AV2 2b and CP-RT act synergistically to allow persistent infection by suppressing RDR6-dependent RNAi, we created the RNA2-st2b clone where we introduced two stop codons 7 and 22 codons after the start codon of ORF2b. AV2-st2b (RNA1, RNA2-st2b and RNA3), AV2-st2b-2st (RNA1, RNA2-st2b and RNA3-2st) and AV2-st2b-mutZF (RNA1, RNA2-st2b and RNA3-mutZF) were inoculated into RDR6i and wild type *N. benthamiana*. Surprisingly, knockout of 2b in the wild type virus or in the CP-RT mutants did not affect virus accumulation in either *N. benthamiana* genotype (SN Fig) suggesting that CP-RT allows persistent AV2 infection by suppressing RDR6-dependent RNA silencing independently of 2b.

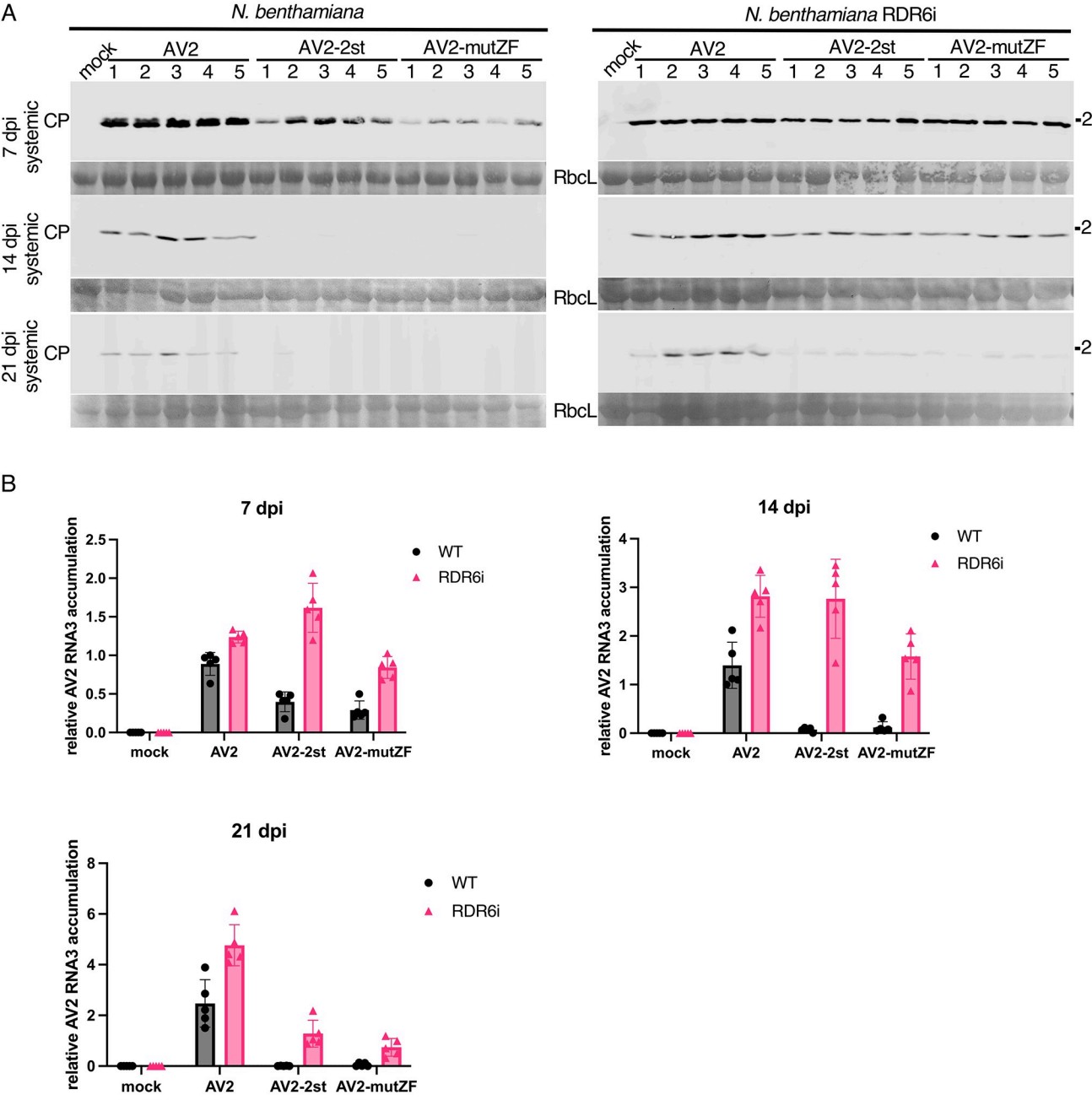

**Fig 8. The different course of infection of AV2, AV2-2st and AV2-mutZF in wild type and RDR6i *N. benthamiana* plants. (A)** Detection of CP by western blot in wild type *N. benthamiana* (left panels) and RDR6i *N. benthamiana* (right panels) infected with AV2, AV2-2st or AV2-mutZF. Samples were collected at 7 dpi from the 2nd upper non-inoculated leaf, and at 14 and 21 dpi from the 3rd upper non-inoculated leaf. Sizes of molecular weight markers are indicated on the right. Ponceau red staining of the large Rubisco subunit (RbcL) was used as a loading control. **(B)** RT-qPCR analysis of RNA3 accumulation at the same time points. *Nb ACT-b* (GI:380505031) was used as a reference gene. Values are expressed in arbitrary units and represent the mean +/− SD ($n = 5$).

## CP-RT is indispensable for persistent infection of *N. benthamiana* shoot apices

It has been demonstrated that symptom recovery from viral infection can result from the induction of RNAi in the shoot apices of infected plants which depends on the activity of host

RDRs [42,53,100,103,106]. To investigate whether CP-RT affects AV2 accumulation in the Shoot Apical Meristem (SAM), longitudinal sections of the top-most flowers and shoot apices from wild type and RDR6i *N. benthamiana* plants that were either mock infected or infected with AV2, AV2-2st or AV2-mutZF were examined by *in situ* hybridisation with digoxigenin-labelled AV2 RNA3 probes on 7 and 14 dpi. In wild type *N. benthamiana* infected with AV2-2st or AV2-mutZF, virus was detected in SAM only at 7 dpi and completely vanished by 14 dpi (Fig 9A). In contrast, in RDR6i *N. benthamiana* infected with AV2-2st or AV2-mutZF, virus was detected in SAM even at 14 dpi although at a lower level compared with AV2-infected plants (Fig 9B). We also analysed virus accumulation in top newly grown tissues of infected plants, in the same experimental set up as was used for *in situ* hybridisation, by RT-qPCR with RNA3-specific primers. Shoot apices of five infected plants at 7 and 14 dpi were pooled together for RNA extraction. The result was consistent with the *in situ* hybridisation (Fig 9C). Thus, AV2, AV2-2st and AV2-mutZF accumulation in SAM corresponds with their accumulation in upper non-inoculated leaves.

### Characterisation of virus-derived siRNAs in AV2 and AV2-2st infected *N. benthamiana*

To confirm that the transient nature of AV2-2st infection is a consequence of the systemic antiviral RNAi response, we analysed virus-derived short-interfering RNAs (vsiRNAs) by high throughput sequencing total siRNA in the top non-inoculated leaves of *N. benthamiana* infected with AV2 or AV2-2st. Samples for RNA extraction were collected at 7 dpi and 14 dpi. Sequencing reads were mapped to the AV2 or AV2-2st virus genomes and the *N. benthamiana* reference genome, and reads per sample counted. The total number of 21 nt reads mapping to a curated set of 128 host miRNAs (described below) per sample was used for normalisation.

A high proportion of virus-mapping reads in both AV2 and AV2-2st infected samples were 21 nt (38–47% of reads) or 22 nt (22–31% of reads) in length (Fig 10A). There was no apparent difference in read length distribution in virus-mapping reads between the AV2 and AV2-2st infected samples. Reads mapping to the virus were evenly distributed between the plus and minus strand for all infected samples (Fig 10B). The proportion of reads mapping to each of the three segments of AV2 was also consistent between the AV2 and AV2-2st infected samples, with many more reads mapping to RNA3 than to RNAs 1 and 2 (Fig 10C, SO Fig). Raw read counts are provided in SD Table.

In all infected samples, after normalisation, virus-mapping read counts at the later time point were significantly higher ($p < 0.05$; Fig 10D; SE Table). Furthermore, there was a clear and significant accumulation of virus-mapping 21, 22 and 24 nt reads, and therefore of vsiR-NAs, in the AV2-2st samples compared to the AV2 samples ($p \leq 1 \times 10^{-4}$; Fig 10D, SE Table). This difference is very apparent at both time points, with an approximately 10-fold increase in 21 nt read counts between AV2-2st and AV2 samples at 7 dpi and a 4-fold increase at 14 dpi (SE Table). It therefore appears that knocking out CP-RT enhances the accumulation of vsiRNAs.

Across mock, AV2 and AV2-2st infected samples, 88–91% of all reads could be mapped to either the host or the virus. For mock infected samples, at both time points, 89–90% of all reads mapped to *N. benthamiana* and 0% to AV2; for AV2 infected samples, 86–89% of reads mapped to the host and 1–4% to the virus; and for AS2-2st infected samples, 73–78% of reads mapped to the host and 12–16% to the virus (Fig 10D, SP Fig). These differences are even more stark if only 21 nt reads are examined, with 5–18% of 21 nt reads mapping to the virus in AS2 infected samples and 42–48% in AS2-2st infected samples (Fig 10D). There was a clear difference in length distribution between the reads mapping to the virus and to the host: in the

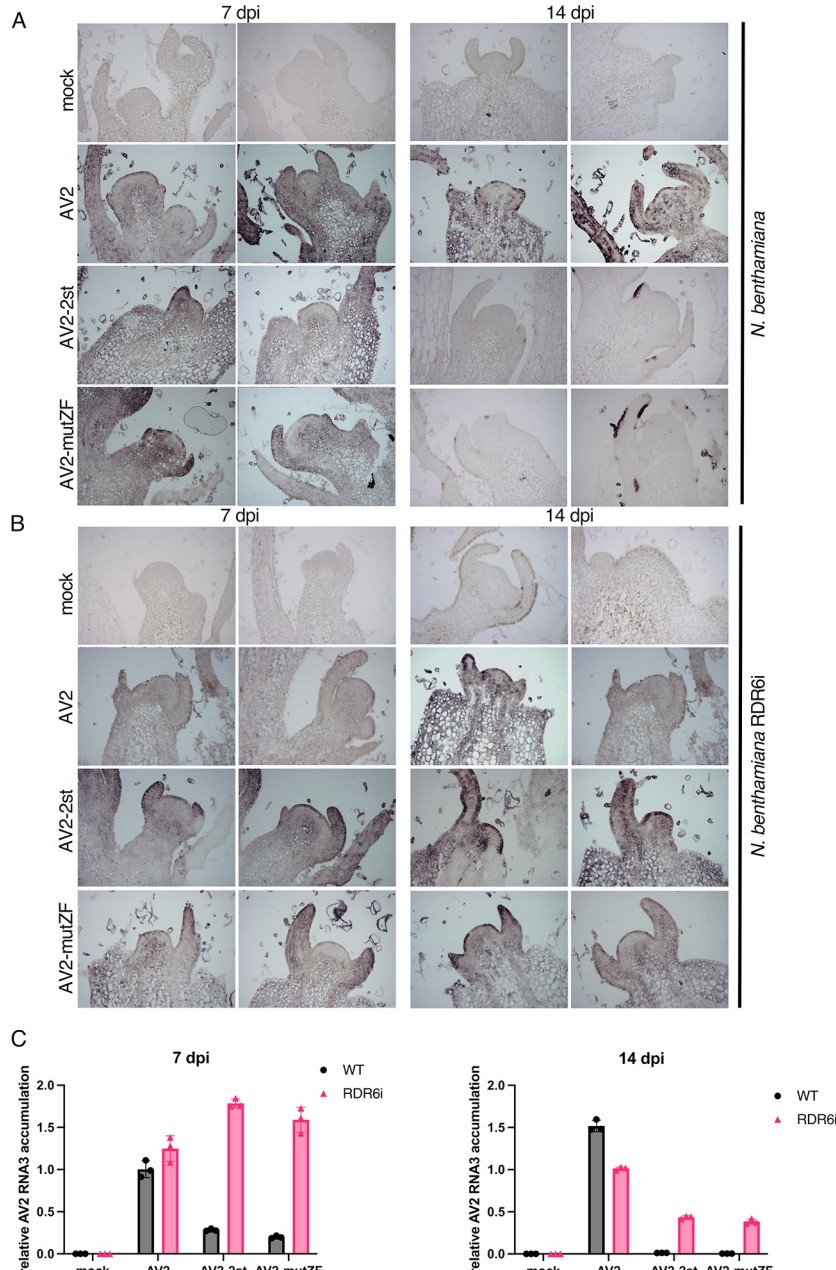

**Fig 9. Detection of RNA3 in longitudinal sections of the top-most shoot apices. (A)** Wild type *N. benthamiana* plants were infected with AV2, AV2-2st or AV2-mutZF, and samples were collected at 7 and 14 dpi. AV2 RNA was detected by *in situ* hybridisation using the digoxigenin (Roche Diagnostics GmbH) labelled *in vitro* transcribed RNA fragment complementary to the AV2 CP ORF. Darker blue-purple areas indicate hybridisation signal indicative of viral RNA. **(B)** As for panel A but using RDR6i transgenic *N. benthamiana* plants. In A and B two representative sections are shown for each treatment at each time-point. **(C)** RT-qPCR analysis of RNA3 accumulation in apical tissues of infected plants. Samples were collected at the same time points as for *in situ* hybridisation. Shoot apices of five infected plants were pooled together for RNA extraction. *Nb ACT-b* (GI:380505031) was used as a reference gene. Values are expressed in arbitrary units and represent the mean +/− SEM of three technical replicates.

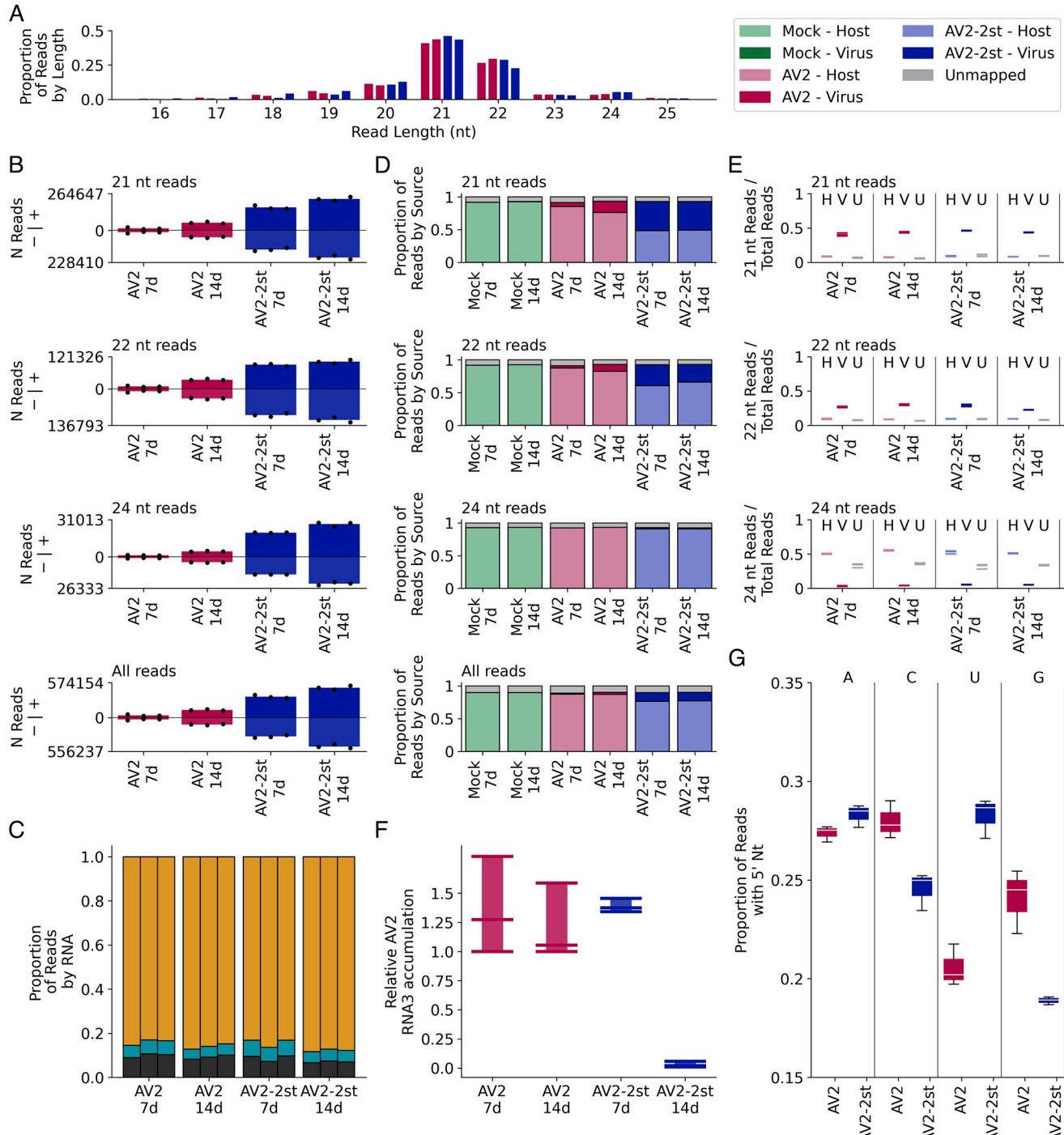

**Fig 10. Characteristics of virus and host small RNAs. (A)** The number of virus mapping reads of each length, divided by the total number of virus mapping reads (summed across replicates for each time point) for the AV2 (pink, bars 1 and 2 in each block) and AV2-2st (blue, bars 3 and 4 in each block) infected samples. **(B)** For 21 nt, 22 nt, 24 nt and all read lengths, the normalised number of reads mapping to the positive (above axis) and negative (below axis) strands of the AV2 genome for AV2 and AV2-2st infected samples, at each time point. Bar heights show the mean for each condition and timepoint; black dots show the individual replicates. **(C)** The number of 21 nt reads mapping to each viral RNA, divided by the number of virus mapping reads in the replicate, separated by replicate and time point, for AV2 and AV2-2st infected samples. RNA1 is shown in black at the bottom of each bar, RNA2 in cyan in the centre, and RNA3 in orange at the top. Other read lengths are shown in SO Fig. **(D)** For 21 nt, 22 nt, 24 nt and all read lengths, the proportion of reads which mapped to the host (bottom section of each bar, light colours), virus (middle section, dark colours) and was unmapped for each sample (top section, grey), separated by treatment and time point, and summed across replicates (see SP(i) Fig for individual replicates). **(E)** The number of host mapping (left, light colours), virus mapping (middle, dark colours) and unmapped (grey) 21 nt, 22 nt and 24 nt reads, divided by the total number of reads mapping to the

same destination (i.e. host, virus or unmapped), for AV2 and AV2-2st infected samples at each time point. Horizontal lines represent individual replicates. **(F)** RT-qPCR analysis of AV2 RNA3 accumulation in the samples used for high throughput siRNA sequencing. *Nb ACT-b* (GI:380505031) was used as a reference gene. Values are expressed in arbitrary units; coloured bars and black error bars represent the mean +/− SEM of three technical replicates (black dots). **(G)** The number of virus mapped 21 nt reads, with each possible 5′-terminal nucleotide for AV2 and AV2-2st infected samples at 14 dpi, divided by the total number of virus mapped 21 nt reads for the same sample and timepoint. See SQ Fig for 22 nt, 24 nt and all reads.

host 7–11% of mapped reads were 21 nt, 7–10% were 22 nt and 28–37% were 24 nt, whereas in the virus these numbers were 38–47% 21 nt, 22–31% 22 nt and 3–6% 24 nt (Fig 10E).

We quantified viral genomic RNAs by RT-qPCR. The amount of AV2 RNA3 was lower in AV2-2st infected plants compared with AV2 infected plants at 14 dpi (Fig 10F). At 14 dpi, the proportion of vsiRNAs with 5′ terminal U residues was increased for 21 nt and 22 nt reads in the AS2-2st infected samples compared to the AS2 infected samples (Fig 10G, SQ Fig). In the absence of CP-RT, the AV2-2st infected samples exhibited a general increase in vsiRNA production along the length of the viral genome (21 nt reads–Fig 11; 22 nt, 24 nt, and all read lengths–SR Fig). Taken together these data indicate that AV2-2st triggers stronger systemic antiviral RNAi than AV2.

An accurate host miRNA annotation was required both for miRNA analysis and to find normalisation factors for viral small RNA quantification. To ensure quality, three sources of host miRNA annotations were combined: those identified by Baksa et al. [107], those provided in the sRNAanno database [108], and a *de novo* ncRNA annotation. 128 high quality miRNAs were selected from this combined set which showed good levels of expression in our samples (for selection criteria see Materials and methods). Per replicate 21 nt read counts for miRNA-mapping reads are shown in SS Fig. Counts of reads mapping to each of the curated miRNAs normalised by the total number of host mapping reads, for each sample and replicate, are available in Online Fig 1 at https://github.com/KatyBrown/ilarvirus_sequencing; there is no apparent global trend–instead miRNA expression appears to be broadly consistent across samples. Counts of 21 nt reads mapping in the poisitive sense to the set of 128 curated miRNAs were used for normalisation and are provided in SF Table.

## Discussion

Through an inspection of their genome sequences, we found that many ilarviruses have a potential readthrough ORF positioned between the CP stop codon and the beginning of the 3′UTR conserved RNA elements. Moreover, where such an insert is present, the CP stop codon context often has features associated with stop codon readthrough. Thus we hypothesised the expression of a C-terminally extended readthrough version of CP, which we named CP-RT. In this study we tested the expression and function of CP-RT using AV2 as a model ilarvirus.

As a start point, we constructed a cDNA infectious clone of AV2 and introduced point mutations to prevent readthrough. CP-RT knockout AV2 was able to replicate and to move cell to cell and long distance in *N. benthamiana*, but this infection was transient, whereas inoculation with wild type AV2 led to persistent infection. Both wild type and mutant AV2 were asymptomatic in *N. benthamiana*. CP-RT was detected *in vitro* upon translation of sgRNA4 in wheatgerm extract and *in planta* in leaves systemically infected by AV2 expressing C-terminally tagged CP-RT. Ribosome profiling also revealed ribosomes translating the RT ORF of wild type but not mutant AV2. An heterologous RT domain from TSV, an ilarvirus belonging to a different subgroup, partially complemented the RT domain of AV2. Thus, CP-RT is produced upon virus infection where it plays an important role in the AV2 life cycle, and its function is evolutionarily conserved between two divergent members of the ilarvirus genus. A key

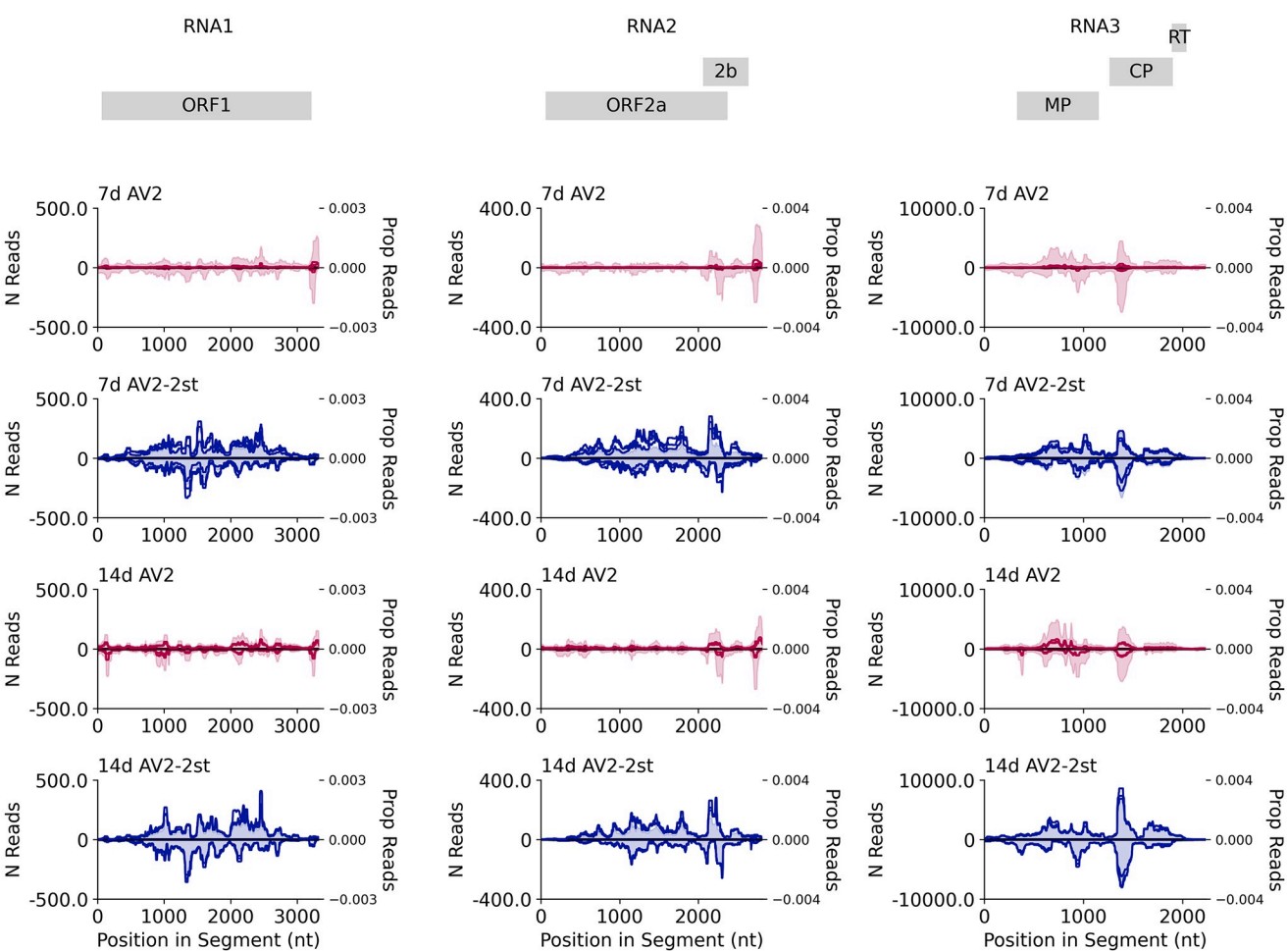

**Fig 11. Distribution of vsiRNA reads on the AV2 genome.** The normalised number of 21 nt reads (shown as one solid line per replicate; left *y* axis) and the proportion of all 21 nt reads mapping to this RNA (shaded and summed across replicates; right *y* axis) spanning each position in the viral genome across RNA1 (left), RNA2 (middle) and RNA3 (right). Both annotations are smoothed with a 100 nt sliding window. AV2 infected samples are shown in red (upper panels) and AV2-2st infected samples in blue (lower panels). Positive-sense reads are shown above the axis and negative-sense reads below the axis. Similar plots for other read lengths (22 nt, 24 nt, all reads) are shown in SR Fig. Note–as discussed previously–reads that map to the conserved 3′ regions may be assigned at random to one of the three segments, potentially leading to an apparent over-abundance in RNAs 1 and 2 due to mismapping of reads that actually derive from the more abundant RNA3.

structural element found in both the AV2 and TSV RT domains is a putative zinc finger. Point mutations disrupting this element led to the attenuation of AV2 and AV2-TSV_RT, thus supporting its functional importance. The mechanistic role of the zinc finger in VSR activity remains undetermined though, interestingly, a number of other plant RNA virus VSRs also contain zinc fingers that in some cases have been shown to be involved in the VSR activity [109,110].

All non-synonymous mutations of the RT domain or even the addition of a tag changed AV2 infection from persistent to transient. This pattern of infection, where the virus can establish systemic infection at first but then fades away and is below the limit of detection in the newly developed leaves, resembles the phenomenon of recovery (given that AV2 is asymptomatic in *N. benthamiana*) which is defined by the appearance of leaves without symptoms and with reduced virus titers following a systemic symptomatic infection [111]. It is widely assumed that recovery is based on the action of RNAi as an antiviral defence [27,111,112].

Therefore, we performed classical transient co-expression assays, with two different types of RNAi inducer, to investigate CP-RT VSR capacity. We found that CP-RT acts as a weak VSR and suppresses RNAi induced by overexpression of single-stranded RNA (ssRNA) but not double-stranded RNA (dsRNA). It is widely assumed that RNAi triggered by overaccumulation of ssRNA (e.g. RNAi of a transgene) relies on the activity of host RDRs and, in *N. benthamiana*, primarily RDR6 [100,113–115]. We found that the attenuation effect of CP-RT mutations is partially compromised in RDR6 knockdown *N. benthamiana*, which suggests that CP-RT suppresses the RDR6-dependent branch of the RNAi pathway. To confirm that the ability of AV2 to induce persistent infection depends on CP-RT VSR activity, we analyzed viral siRNA accumulation in *N. benthamiana* infected with AV2 and AV2-2st at two time points by high throughput sequencing of siRNAs. 21, 22 and 24 nt siRNAs serve as a hallmark of RNAi. They are produced by host DCL proteins and can be amplified by host RDRs to increase the efficiency of RNAi [116,117]. We found that the levels of virus-derived siRNAs were much lower for AV2 than for AV2-2st and conversely the level of genomic RNA3 was higher for AV2 than for AV2-2st, indicating that AV2-2st induces stronger RNAi than AV2. We therefore conclude that the VSR activity of CP-RT is indispensable for AV2 persistence.

Argonaute (AGO) proteins are the key components of the RNAi response in plants. AGOs incorporate siRNAs into RNAi effector complexes. *Arabidopsis* encodes 10 AGOs of which AGO1, AGO2, and to some extent AGO5 are the most important players in virus-induced RNAi [118–120]. The 5′ nucleotide of siRNA determines its targeting into different AGO complexes, where AGO1 preferentially binds to siRNAs with a 5′-terminal U residue [121]. Interestingly, in our screen we found that the proportion of vsiRNAs with 5′ terminal U residues was significantly increased for 21 nt and 22 nt reads in AV2-2st infected compared to AV2 infected samples.

The silencing function was also tentatively supported by the ribosome profiling data, where one possible explanation for the 5′ excess density of ribosome footprints on AV2-2st viral RNAs but not on host mRNA or AV2 viral RNAs could be cleavage of viral RNAs by RISC in AV2-2st infections, but suppression of this response in AV2 infections. Cleavage of actively translating viral RNAs would prevent ribosomes from reaching 3′ regions, leading to an excess density of ribosomes in 5′ regions compared to 3′ regions. However, this observation is based on a single Ribo-seq replicate for each virus so needs to be treated with caution.

Recovery from a viral infection can result from the induction of RNAi in the shoot apices of infected plants, which in its turn depends on virus entry into SAM and RDR6 activity [42,52,53,100,103,106,122,123], although in some cases recovery can appear independently of RNAi as a result of transcriptional reprogramming in SAM [124]. It has been shown that 16K, the weak VSR of tobacco rattle virus (TRV), allows transient infection of SAM which leads to recovery and long-term virus exclusion, and it was suggested that the weakness of a VSR is a crucial factor ensuring SAM transient infection [55]. For cucumber mosaic virus, mutations in CP which increased the silencing suppression activity of 2b were found to change the virus accumulation pattern in SAM from transient to persistent and consequently abolished recovery in wild type *N. benthamiana* but not in RDR6 or SGS3 mutants [125]. A turnip mosaic virus variant with point mutations decreasing the VSR activity of HC-Pro was able to infect SAM while the wild type virus was excluded from SAM [126]. It was also demonstrated that HC-Pro of another potyvirus, sugarcane mosaic virus, downregulates the accumulation of host RDR6 mRNA [127]. It seems that there is a certain level of silencing suppression which allows transient virus entry into the meristems that then leads to subsequent recovery where key players are RDR6, from the host side, and a VSR from the virus side. The recovery-like phenotype induced by CP-RT mutant AV2 might be explained by modulation in the VSR activity of CP-RT leading to a change in the AV2 accumulation pattern in SAM. To investigate this, we

analysed virus distribution and accumulation in the shoot apices of wild type and RDR6 knockdown *N. benthamiana* infected with AV2, AV2-2st or AV2-mutZF. We found that virus accumulation in upper non-inoculated leaves corresponded to that in SAM: AV2 CP-RT mutants were only able to transiently infect SAM of wild type plants, whereas they could persistently infect SAM of plants where RDR6 expression was knocked down. Further investigation is needed to find out whether the recovery-like phenotype of infection with CP-RT mutants is a consequence of transient invasion of SAM or whether both are manifestations of the same process.

It was previously shown that AV2 2b is a VSR that suppresses systemic but not local RNAi [63]. Some viruses encode more than one VSR that target different steps of the RNAi pathway [64–68] or act cooperatively [69]. This strategy allows more sophisticated regulation of RNAi. Persistent asymptomatic infection can be considered as the best-adapted virus-host interaction [128]. Part of the symptoms occurring during acute plant virus infection can be attributed to dysregulation of host gene expression due to the suppression of RNAi by VSRs [129,117]. Having two weak VSRs acting on different steps of the RNAi pathway might be beneficial to maintain a certain level of RNAi controlling AV2 accumulation and preventing symptom appearance, facilitating persistent infection, meristem invasion and vertical transmission. In asparagus, AV2 does not induce visible symptoms but is thought to be the cause of the Asparagus decline syndrome [25]. Despite a similar infection phenotype, further study is needed to characterise the role of CP-RT in AV2 pathogenesis in its natural host.

Although our experimental studies were limited to AV2, our bioinformatic analysis extended across all ilarviruses with RNA3 3′UTR sequence data. Our analysis indicated that an RT domain–of some form–is present in many but not all ilarvirus species. Notably (using the classification in Table 5.3 of Ref. [22]), an RT domain was predicted in all members of subgroups 1, 2 and 4, but none of the members of subgroup 3 except for apple mosaic virus where an independently evolved RT domain was predicted in one lineage of apple mosaic virus. The insert was particularly long in Fragaria chiloensis latent virus, and in this one case the potential for an ORF downstream of CP has been previously noted [130]; although in that paper it was flagged as a shorter putative AUG-initiated ORF, the presence of a UAG CAA UUA Type I readthrough motif at the CP stop codon is more consistent with readthrough expression. It may be that the 2b and RT ORFs were both not present in the last common ancestor of ilarviruses, thus explaining why–even though advantageous–they are not present in all members of the genus.

As noted in the Introduction, ilarvirus 3′UTRs contain multiple repeats of an AUGC motif that binds CP to promote translation of the non-polyadenylated viral RNAs. As AUGC includes a UGC triplet (Cys codon), it is tempting to speculate that ilarivirus zinc finger or other Cys-rich RT domains may have originated via duplication of 3′UTR sequences containing a number of these AUGC motifs such that, by chance, multiple UGCs were in frame with the CP ORF. Once repurposed as a coding sequence, these AUGCs could be free to mutate (in our AV2 RT sequence, none of the eight Cys codons coincide with an AUGC tetranucleotide). There are, nonetheless, a small number of ilarvirus species where the putative RT domain definitely does not encode a zinc finger, and so our results with AV2 cannot be extrapolated to all ilarvirus species with a putative RT domain. Some cases (particularly cluster 6b) almost certainly had an independent evolutionary origin from the AV2 RT domain and therefore likely also have different functions. As seen with other viruses, the C-terminus of a CP protein may be a convenient location to append an extension domain either via frameshifting [131] or readthrough [1]. We currently do not know whether or not ilarvirus CP-RT proteins are incorporated into virions and, if so, whether it is functional or accidental. Previous SDS-PAGE (e.g. Ref. [132]) and anti-CP western blotting (e.g. Refs [133,134]) of virion proteins for various

ilarviruses suggest that it is not or, if it is, it is only at very low levels. However our own work indicates that the CP-RT:CP expression ratio is also extremely low, at least for AV2. It is possible that maintaining a low RT efficiency is important to prevent CP-RT interference with capsid form or function.

In conclusion, our work has revealed a new ilarvirus protein and a new example of programmed stop codon readthrough, and provides new insights into how plant RNA viruses can maintain persistent infection in the face of host antiviral RNAi.

## Materials and methods

### Comparative genomics

Ilarvirus sequences were downloaded from GenBank (most recently on 11 November 2022) by searching the NCBI nucleotide database with the query "txid12316[Organism:exp]". In total 2722 sequences were downloaded including 74 from RefSeq; 25 of the 74 RefSeqs were for RNA3. Patent sequence records were removed (six sequences). To identify CP-encoding sequences (i.e. RNA3s), we used blastx [73] using a database of the RefSeq CP amino acid sequences as the subject, a word size of 2, and an e-value threshold of 0.01. We discarded any sequences with < 60 nt downstream of the CP stop codon. Then we identified the maximal start-codon-to-stop-codon CP ORF lengths in the remaining sequences; there was a gap in these lengths between 498 nt and 606 nt and, given that the CP ORF length in the 25 RNA3 RefSeqs ranged from 612 to 741 nt, we discarded all sequences where the maximal start-codon-to-stop-codon CP ORF length was less than 600 nt. An exception was made for KY488189 which has an internal stop codon in the CP ORF (presumably a sequencing error/defective sequence). At this stage, 437 CP-encoding sequences remained.

In some cases, the maximal start-codon-to-stop-codon ORF may start at an AUG codon upstream of the actual utilised AUG codon (e.g. in some sequences there is an in-frame AUG upstream of the utilised CP AUG that is not however present on the sgRNA4 transcript). To avoid this issue affecting our BLASTCLUST clustering step (below), we aligned the CP amino acid sequences with MUSCLE [135], manually inspected the CP alignment and, where possible, truncated such spurious N-terminal extensions. We then clustered the 437 nucleotide sequences based on their encoded CP amino acid sequences by using BLASTCLUST [73] (a single-linkage BLAST-based clustering algorithm) with parameters -L 0.90 -b T -S 80 (i.e. 90% coverage, 80% amino acid identity threshold). This resulted in 37 clusters of which 23 contained a RefSeq (two clusters each contained two RefSeqs). We split the two clusters that contained two RefSeqs into subclusters (8a/8b and 17a/17b), one for each RefSeq. We also split clusters 5 and 6 into 5a/5b and 6a/6b due to substantial differences in RT features between subclusters (see SC Data).

In order to determine for each cluster whether there was an RNA3-specific insert (putative RT domain) between the CP stop codon and the beginning of the conserved 3′UTR, we needed to identify matched RNA1 and RNA2 sequences for each cluster. For clusters with an RNA3 RefSeq representative, there were usually matched RefSeqs for RNAs 1 and 2. For the other 15 clusters, where possible we chose an RNA3 reference sequence for the cluster that had matched, complete RNA1 and RNA2 sequences; in a few cases this was not possible. See SC Data for a list of all RNA3 clusters, the chosen reference sequences and matched RNA1 and RNA2 reference sequences where available, and accession numbers for all the non-reference RNA3 sequences in each cluster. Note that reference sequence NC_008706 was 'corrected' via insertion of an 'A' 9 nt downstream of the CP stop codon; this correction was supported by the other four sequences in the cluster (see cluster 11 of SC Data for details).

Coding potential across RNA3 was assessed using MLOGD [74]. For subgroup 1 we used the RNA3 reference sequences OL539725, MG800791, OM323991, OL472062, NC_022129, NC_040394, NC_011555, NC_008706, OL584350, NC_003845 and NC_027930 whereas for subgroup 2 we used the RNA3 reference sequences OL472059, NC_003546, NC_003835, NC_039076, NC_003810, NC_011807, NC_009536 and NC_003570 (see Fig 2). Although NC_005854 falls within the subgroup 1 clade based on CP phylogeny (Fig 2), it was excluded from the MLOGD analysis since its RT domain differs substantially from the RT domains of other sequences in this clade (SF Fig; SC Data–cluster 12). Alignments were generated as described in [136]. Briefly, for each subgroup, each individual RNA3 sequence was aligned to a chosen primary reference sequence (NC_003845 [tobacco streak virus] and NC_011807 [asparagus virus 2], respectively) using code2aln version 1.2 [137], and mapped to the primary reference sequence coordinates by removing alignment positions that contained a gap character in the primary reference sequence. These pairwise alignments were combined to give multiple sequence alignments which were analysed with MLOGD using a 40-codon sliding window and a 5-codon step size. For each of the three reading frames, within each window the null model is that the sequence is non-coding whereas the alternative model is that the sequence is coding in the given reading frame.

Using the reference sequences for each cluster, for each of RNAs 1, 2 and 3 we extracted amino acid sequences for ORFs 1, 2a, MP and CP. For each protein, we aligned the amino acid sequences with MUSCLE v3.8.31 [135]. Maximum likelihood phylogenetic trees were estimated using the Bayesian Markov chain Monte Carlo method implemented in MrBayes v3.2.3 [138], sampling across the default set of fixed amino acid rate matrices, with 1,000,000 generations, discarding the first 25% as burn-in (other parameters were left at defaults). Trees were visualized with FigTree v1.4.2 (http://tree.bio.ed.ac.uk/software/figtree/).

To search for potential homologues or shared motifs, the sequences of putative RT domains were queried with HHpred [77]. Queries were performed through the HHpred webserver on 20 October 2023, using the subject databases PDB_mmCIF70_18Jun and Pfam-A_v36.

## Plant materials and growth conditions

*Nicotiana benthamiana* Domin. plants (Lab accession RA-4) [139] were grown under a 16 h photoperiod under 200 μmol m$^{-2}$ s$^{-1}$ of photosynthetically active radiation at 22°C and 60% humidity in a growth room (Conviron, Winnipeg, MB Canada). The transgenic *N. benthamiana* RDR6i line used in this study was described previously [105].

## Plasmid construction

To construct the AV2 cDNA clone, complete sequences of the three genomic segments of AV2 were obtained by 5′- and 3′-RACE with the SMARTer RACE cDNA amplification kit (Clontech) using infected *Nicotiana occidentalis* leaf material (DSMZ, Germany). Using 5′ RACE reaction as a template, each genomic segment was amplified as two PCR products and cloned in the pDIVA backbone [79] using the NEBuilder HiFi DNA Assembly kit (NEB). The CP-RT ORF (without viral UTRs) was amplified with specific primers followed by insertion of the resulting fragment into the pBin61 plasmid. To construct the AV2 sgRNA4 clone under a T7 promoter, sgRNA4 was amplified with primers specific to its 5′ end (obtained by 5′-RACE) and 3′ end (obtained by 3′-RACE of AV2 RNA3), followed by insertion of the resulting fragment into the BlucB plasmid (kindly provided by Prof. W. A. Miller). To construct AV2 mutants and all other plasmids for this study, standard recombinant DNA procedures were performed by use of a combination of PCRs, site-directed mutagenesis, swapping of restriction fragments and Gibson assembly [140].

### Transient expression and virus inoculation

RNAi suppression assays were carried out as described by Yelina et al. [141] with minor modifications. The pBin-GFP reporter vector (or pBin-GFP along with pBin-dsGF) was co-expressed with plasmids expressing the indicated proteins or with empty vector by agroinfiltration of fully developed leaves of 3-week-old *N. benthamiana*. Samples were collected at 4 dpi. For virus infection, agrobacterium strains carrying pDIVA-AV2RNA1, pDIVA-AV2RNA2, pDIVA-AV2RNA3 (or its mutant version) and pDIVA-CP-RT were mixed and infiltrated into two leaves on each plant of 10 d old *N. benthamiana*. All agrobacterium strains were adjusted to a final optical density (OD$^{600}$) of 0.2.

### SDS-PAGE and immunoblotting

Protein extracts from *N. benthamiana* leaves were prepared from leaf samples (40 mg, fresh weight). Leaf discs were ground and homogenised in 80 μL 250 mM Tris-HCl, pH 7.8, mixed with 40 μL extraction buffer (75 mM Tris-HCl, pH 6.8, 9 M urea, 4.3% SDS, and 7.5% 2-mercaptoethanol), heated at 95°C for 5 min, and centrifuged (3 min at 13,000 *g*). Aliquots of the supernatant (10 μL) were separated by SDS-PAGE on 12% gels. After electrophoresis, proteins were transferred to a 0.2 μm nitrocellulose membrane (Bio-Rad) and subjected to immunoblot analysis with the relevant primary and secondary antibodies (anti HA rabbit polyclonal, Abcam ab20084; anti AV2 CP rabbit polyclonal, DSMZ RT-0953-0645/1; anti c-Myc mouse monoclonal, Sigma M4439; anti-mouse IgG goat, LI-COR 926–32210; anti-rabbit IgG donkey, LI-COR 926–68073). Membranes were imaged using an Odyssey CLx imaging system (LI-COR).

### Total RNA extraction and northern blotting

100–200 mg of deep-frozen leaf material were manually ground in liquid nitrogen. Total RNA was extracted using TRI-Reagent (Sigma-Aldrich) according to the manufacturer's instructions and solubilised in sterile water. High molecular mass RNA was precipitated with an equal volume of 4 M LiCl. For analysis of siRNAs, 10 μg of low molecular mass RNA fraction was separated in a 15 % polyacrylamide gel containing 8 M urea, transferred to a nylon membrane (Hybond-N, Amersham Biosciences AB), cross-linked with UV light, prehybridised in PerfectHyb Plus buffer (Sigma-Aldrich) at 42°C for 1 h, and hybridised overnight with the Cy5 labelled, fragmented negative-sense T7 transcript of the *gfp* gene. The Cy5 probe was generated using the HighYield T7 Cy5 RNA Labelling Kit (Jena Bioscience) with a T7-*gfp* PCR product as template. After hybridisation and washing, membranes were imaged using an Odyssey CLx imaging system (LI-COR).

### Library preparation for siRNA sequencing

10 μg of low molecular mass RNA fraction was separated by urea-PAGE. RNA bands within the size range 18–30 nt were acrylamide gel-purified. All library amplicons were constructed using a small RNA cloning strategy [142] and sequenced (single-end, 75 nt) using the NextSeq 500 platform (Illumina).

### Library preparation for RNA-seq

5 μg of high molecular mass RNA was subjected to alkaline fragmentation and separated by urea-PAGE. RNA bands within the size range 18–30 nt were acrylamide gel-purified. All library amplicons were constructed using a small RNA cloning strategy [142]. Libraries were

generated for three separate plants for each virus Libraries were sequenced (single-end, 75 nt) using an NextSeq 500 platform (Illumina).

RNA-Seq analysis was performed as described in Irigoyen et al. [91]. Adaptor sequences were trimmed using the FASTX-Toolkit v0.0.13 (http://hannonlab.cshl.edu/fastx_toolkit), adaptor-only reads, non-clipped reads, and trimmed reads shorter than 25 nt were discarded. The mean length of the retained reads was in the range 47.0–52.1 across the six libraries. Reads were mapped to the AV2 or AV2-2st infectious clone sequences using bowtie v0.12.9 [143], with parameters -v 2—best (i.e. maximum 2 mismatches, report best match). Contig assemblies were performed on RNA-seq datasets with the SPAdes genome assembler v3.15.5 [84] in rnaviralSPAdes mode [85], with the default settings for single end reads.

### Ribosome profiling

For each library preparation 500 mg of leaf material was flash frozen and pulverised in liquid nitrogen with 1.5 mL of cold lysis buffer [89]. Subsequent stages were performed as described in previously [89,91]. The Ribo-Zero Gold rRNA removal kit (Illumina) was used for rRNA depletion. Libraries were sequenced (single-end, 75 nt) using a NextSeq 500 platform (Illumina).

Ribo-Seq analysis was performed as described in Irigoyen et al. [91]; Ribo-seq pipeline scripts are available at www.firthlab.path.cam.ac.uk/index.html and at https://github.com/AndrewFirth12/RiboseqAnalysis. Adaptor sequences were trimmed using the FASTX-Toolkit v0.0.13 (http://hannonlab.cshl.edu/fastx_toolkit) and adaptor-only reads, non-clipped reads, and trimmed reads shorter than 20 nt were discarded. Reads were mapped to host (*N. benthamiana*) and virus (AV2 or AV2-2st) RNA using bowtie v0.12.9 [143], with parameters -v 2—best (i.e. maximum 2 mismatches, report best match). Mapping was performed in the following order: host rRNA, virus RNA, host mRNA. The virus transcript database comprised the sequences of the three segments of our AV2 or AV2-2st infectious clones as appropriate. For the host rRNA database we used GenBank accessions AJ131166.1, AJ131167.1, FJ217346.1 and KP824745.1. For the host mRNA database we combined files Niben101_annotation.transcripts.fasta and Niben.genome.v0.4.4.transcripts.fasta from Ref. [144]; then we selected all transcripts containing an AUG-codon-to-stop-codon ORF of at least 900 nt and annotated the longest such ORF in the transcript as the putative coding sequence. It should be noted that this *ad hoc* method of CDS annotation may occasionally flag up the incorrect initiation site, especially if transcripts are incomplete. It will also omit mRNAs with CDSs less than 900 nt in length. However, our aim in designating a host mRNA database was simply to use it to assess Ribo-seq quality not to assess host mRNA gene expression, and (as illustrated in SJ(ii) Fig) our mRNA database proved to be perfectly adequate for this purpose.

For Fig 5D, SK and SL Figs, a +12 nt offset was applied to the RPF 5′ end positions to give the approximate ribosomal P-site positions. To calculate the read length and phasing distributions of host and virus RPFs (Fig 5A and 5B, and SJ(i) Fig), only RPFs whose 5′ end (+12 nt offset) mapped between the 13th nucleotide from the beginning and the 18th nucleotide from the end of coding sequences were counted, thus avoiding RPFs near initiation and termination sites; the dual-coding region where ORF2b overlaps ORF2a was also excluded. Histograms of host RPF positions relative to initiation and termination sites (Fig 5C and SJ(ii) Fig; no +12 nt offset applied here) were derived from RPFs mapping to host mRNAs with annotated 5′ and 3′ UTRs each of at least 60 nt in length, and CDS of at least 1200 nt.

### Quantitative reverse transcription-coupled PCR (RT-qPCR)

Total plant RNA was treated with DNase I (Promega) according to the manufacturer's recommendations. 500 ng of total RNA were reverse-transcribed using the QuantiTect Reverse

transcription kit (Qiagen). RT-qPCR was performed in triplicate using SsoFast EvaGreen Supermix (Bio-Rad) in a ViiA 7 Real-time PCR system (Applied Biosystems) for 40 cycles with two steps per cycle. Transcript levels were normalised to that of *Actin* (*ACT-b* GI: 380505031).

### *In vitro* translation

BlucB-sgRNA4 and its derivatives were linearised with *Acc65*I just downstream of the authentic sgRNA4 3′ end, and capped run-off transcripts were generated using T7 RNA polymerase (MEGAscript T7 transcripton kit, Thermo Fisher). mRNAs were translated in wheatgerm extract (Promega) supplemented with 20 mM amino acids (lacking methionine) and 0.2 MBq [$^{35}$S]-methionine. Reactions were incubated for 1 h at 30˚C and stopped by the addition of an equal volume of RNaseA 100 μg/ml in 10 mM EDTA. Proteins were resolved by 12% SDS-PAGE and dried gels were exposed to a Cyclone Plus Storage Phosphor Screen (PerkinElmer). The screen was scanned using a Typhoon TRIO Variable Mode Imager (GE Healthcare).

### *In situ* hybridization

Shoot and floral apices were collected from infected or mock infected *N. benthamiana*, embedded in wax, sectioned, and *in situ* hybridised as described previously [145,146]. AV2 RNA was detected with the digoxigenin (Roche Diagnostics GmbH) labelled *in vitro* transcribed RNA fragment reverse complementary to the AV2 CP ORF, and then detected with anti-digoxigenin antibody conjugated to alkaline phosphatase (Roche Diagnostics GmbH) and Western Blue stabilised substrate (Promega).

### Computational analysis of vsiRNAs

High throughput sequencing reads were trimmed for adapters using Trim Galore v0.6.7 (https://www.bioinformatics.babraham.ac.uk/projects/trim_galore/), with the default settings plus a minimum read length of 10 nt and maximum read length of 50 nt. More than 97% of reads across all samples contained adapters, as expected for small RNA sequencing. Reads were then mapped to the host genome (Nbv0.5) [147] using bowtie v1.3.1 (Langmead et al., 2009) allowing no mismatches (-v 0) and with optimal alignment in terms of number of mismatches and quality (—best). Reads were also mapped to the AV2 or AV2-2st genome (as appropriate) with bowtie using the same settings. Reads were separated by length and strand using the pysam v0.19.0 (https://github.com/pysam-developers/pysam) wrapper for samtools v1.6 [148]. The number of reads spanning each position in the viral genome was counted using samtools v1.6 mpileup [148] with a maximum depth (-d) of 500,000 and quality filtering and adjustment disabled (-Q 0, -q 0, -B -C 0). Read counts were normalised using the number of 21 nt reads mapping to host miRNAs, as described below.

### Micro RNA annotation

To ensure quality, three sources of host miRNA annotations were combined: those identified by Baksa et al. (2015) [107], those provided in the sRNAanno database [108], and a *de novo* ncRNA annotation. For the Baksa et al. and sRNAanno miRNAs, coordinates on Nbv0.5 were identified using BLASTN v2.12.0 [73]. The *de novo* ncRNA annotation was generated using Infernal v1.1.2 [149] using annotations from Rfam release 14.7 [150] and the options—cut_ga and—rfam, as recommended in the Rfam documentation. These sources were combined to generate a confident set of 128 miRNAs present in our samples by taking regions which, in at least 3 of our replicates, had ≥20 mapped reads, at least 50% of which were 21 nt in length,

with at least twice as many reads mapping to one strand as to the other, and at least twice as many reads mapping to the putative miRNA than the putative miRNA\*. The strand of the miRNA was always taken to be the strand where the majority of reads mapped, and the miRNA position to be the 21 nt with the most reads within the precursor. This miRNA annotation is not intended to be comprehensive, but rather to represent a subset of high quality miRNA positions. A bed file containing this annotation (relative to Nbv0.5) is available at https://github.com/KatyBrown/ilarvirus_sequencing.

Read counts per miRNA represent the total number of reads of each length mapping fully within the miRNA, generated using bedtools intersect v2.30.0 [151] of all host mapping reads against the bed file above with the option -f 1 (100% overlap only). Normalisation factors for viral read counts were generated using the 21 nt reads of this type.

## Supporting information

**S1 Data. Supplementary Information File combining all supplementary figures, supplementary tables, and supplementary data.**
(PDF)

## Acknowledgments

We thank the DNA Sequencing Facility (Department of Biochemistry, University of Cambridge) and Cambridge Genomic Services (Department of Pathology, University of Cambridge) for high throughput sequencing. For the purpose of open access, the authors have applied a CC BY public copyright licence to any Author Accepted Manuscript version arising from this submission.

## Author Contributions

**Conceptualization:** Nina Lukhovitskaya, Katherine Brown, John P. Carr, Andrew E. Firth.

**Data curation:** Katherine Brown, Andrew E. Firth.

**Formal analysis:** Nina Lukhovitskaya, Katherine Brown, Andrew E. Firth.

**Funding acquisition:** John P. Carr, Andrew E. Firth.

**Investigation:** Nina Lukhovitskaya, Katherine Brown, Lei Hua, Andrew E. Firth.

**Methodology:** Nina Lukhovitskaya, Katherine Brown, Lei Hua, Andrew E. Firth.

**Project administration:** Andrew E. Firth.

**Resources:** Adrienne E. Pate.

**Validation:** Nina Lukhovitskaya, Katherine Brown.

**Visualization:** Nina Lukhovitskaya.

**Writing – original draft:** Nina Lukhovitskaya, Katherine Brown, Andrew E. Firth.

**Writing – review & editing:** Nina Lukhovitskaya, John P. Carr, Andrew E. Firth.

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
