## [Decision Letter · Decision Letter 0]

9 Mar 2024

Dear Lukhovitskaya,

Thank you very much for submitting your manuscript "A novel ilarvirus protein is expressed via stop codon readthrough and suppresses RDR6-dependent RNA silencing" for consideration at PLOS Pathogens. As with all papers reviewed by the journal, your manuscript was reviewed by members of the editorial board and by three independent reviewers. The reviewers appreciated the attention to an important topic. Based on the reviews, we are likely to accept this manuscript for publication, providing that you modify the manuscript according to the review recommendations.

Sincerely,

Aiming Wang, Ph.D

Academic Editor

PLOS Pathogens

Shou-Wei Ding

Section Editor

PLOS Pathogens

Michael Malim

Editor-in-Chief

PLOS Pathogens

orcid.org/0000-0002-7699-2064

Reviewer Comments (if any, and for reference):

Reviewer's Responses to Questions

**Part I - Summary**

Reviewer #1: (No Response)

Reviewer #2: The manuscript authored by Lukhovitskaya et al reports a novel readthrough protein (CP-RT) shared by a variety of ilarviruses, and, by using the pathosystem of asparagus virus 2 and Nicotiana benthamiana, discloses its functional role in suppression of host RNA silencing to facilitate viral persistent infection. I appreciate the thorough bioinformatic analysis in defining a putative readthrough domain immediately followed by CP ORF among many subgroups of ilarviruses. The authors adopted a series of approaches, e.g., reverse genetic, wheatgerm extract-based in vitro translation, and ribosome profiling, to demonstrate an actual existence of the readthrough event and its essentiality in supporting viral persistent infection. Along with a hypothesis that RNAi might possibly contribute to the failure of viral persistence / recovery-like phenotype, the authors corroborated a counter-defense role of the readthrough protein likely via targeting the RDR6-involved step in RNAi pathway. Overall, the bioinformatic analysis and experimental design and methods are appropriate, and the obtained results support the conclusions. The novel findings greatly enhance our understanding on the fundamental aspects of ilarvirus biology. However, I still have several comments that need to be properly addressed before its publication, see my concerns in Part II and III.

Reviewer #3: This manuscript presents an extensive investigation into a new protein variant of ilarviruses. These viral genomes are compact and maximizing use of its capacity is assumed. In this study, a novel short extension of the CP by read-through is bioinformatically predicted and extensively proved both to exist in native infections as well as contributing to infection by RNA silencing mechanisms enabling the virus to persist in meristems. Furthermore, demonstrating that the same RT domain from another ilarvirus can complement broadens the context nicely. The work is truly substantial, methodology is top-level and the resulting claims are fully justified. In fact, the only “weakness” I can identify is that the paper is so heavy, i.e. almost 100 pages supplementary. There are some typos and small things in the text, but the lack of line numbering made it too tedious for me to point out such minor faults.

Altogether, I see no need for additional experimentation with the confession that I have limited detailed expertise to evaluate technicalities regarding the sequencing analyses. I think the study has taken the best methods to evidence the predicted RT and its functional meaning. One interesting point for biological relevance would be whether the presence of the RT controls vertical transmission of the virus.

**Part II – Major Issues: Key Experiments Required for Acceptance**

Reviewer #1: (No Response)

Reviewer #2: Amongst different subgroups of ilarviruses, the CP stop codon and surrounding motif for a putative readthrough event is present. In line with this, the “CP-RT” protein was detected by in vitro translation assay, immunoblotting analysis with an antibody against AV2 CP or Myc/HA tag fused at the C-terminus of RT. I believe that the discovery of the readthrough event is a breakthrough in our understanding the fundamental biology of ilarvirus. To further consolidate the conclusion, I suggest following experiments for considerations: 1. Purification of Myc-tagged CP-RT for mass spectrometry (MS) analysis, and thus, a portion of the peptide sequences from MS analysis are expected to correspond with RT sequence. 2. Based on the authors’ analysis, the existence of RT domain is commonly shared by a large number of ilarviruses. It should be a real situation in ilarviral infection in nature. The experimental data about the presence of CP-RT is lacking. Performing immunoblotting analysis of AV2 CP-RT using CP antibody is suggested.

Reviewer #3: None needed for acceptance, vertical transmission regulation by RT appears as a relevant biological possibility.

**Part III – Minor Issues: Editorial and Data Presentation Modifications**

Reviewer #1: (No Response)

Reviewer #2: 1. As shown in Figure 2, for many ilarviruses, they encode both 2b and a potential CP-RT to function in counteracting host RNA silencing. The authors should discuss the functional differences between them in detail. Some ilarviruses have a single 2b (Apple ilarvirus 2), and some contain neither 2b nor CP-RT (Apple mosaic virus - 6a, Blueberry shock virus, etc.). Why are viruses in the same genus so different in viral suppressors of RNA silencing?

2. It is curious that most CP-RT proteins in ilarvirus consist of zinc finger motif, and the motif is associated with RSS activity and viral persistent infection. The zinc finger in viral suppressors of RNA silencing is commonly shared by other viruses in different genera? It is whether or not CP-RTs serve as transcription factors in regulating the expression of RDR6’s expression to affect RNAi pathway in an indirect manner? Does the CP-RTs have a nucleus-localized signal?

3. In Page 12 “we were unable to see a clear unambiguous effect of …………on gfp siRNA accumulation (Figure 7D)”. The suppression of RNA silencing is usually accompanied with a less accumulated vsiRNA? Please clarify this point.

Reviewer #3: The text is somewhat heavy, especially the cross-referencing to the table in the first results section. It makes it hard to grasp at a general level so I therefore suggest to simplify and clearly lift out conclusions to the extent possible without compromising scientific quality of course.

PLOS authors have the option to publish the peer review history of their article (what does this mean?). If published, this will include your full peer review and any attached files.

Reviewer #1: No

Reviewer #2: No

Reviewer #3: No

Figure Files:

Data Requirements:

Reproducibility:

References:

---

## [Editor Report · Decision Letter 1]

3 May 2024

Dear Lukhovitskaya,

We are pleased to inform you that your manuscript 'A novel ilarvirus protein is expressed via stop codon readthrough and suppresses RDR6-dependent RNA silencing' has been provisionally accepted for publication in PLOS Pathogens.

Best regards,

Aiming Wang, Ph.D

Academic Editor

PLOS Pathogens

Shou-Wei Ding

Section Editor

PLOS Pathogens

Michael Malim

Editor-in-Chief

PLOS Pathogens

orcid.org/0000-0002-7699-2064
---

## [Editor Report · Acceptance letter]

23 May 2024

Dear Dr Lukhovitskaya,

We are delighted to inform you that your manuscript, "A novel ilarvirus protein CP-RT is expressed via stop codon readthrough and suppresses RDR6-dependent RNA silencing," has been formally accepted for publication in PLOS Pathogens.

Best regards,

Michael Malim

Editor-in-Chief

PLOS Pathogens

orcid.org/0000-0002-7699-2064